# Assessing the impact of large volcanic eruptions of the Last Millennium on Australian rainfall regimes

Stephanie A. P. Blake[a,b], Sophie C. Lewis[b,c], Allegra N. LeGrande[d] and Ron L. Miller[d]

[a] Climate Change Research Centre, University of New South Wales, Sydney, UNSW, Australia
[b] ARC Centre of Excellence for Climate System Science
[c] Fenner School of Environment and Society, The Australian National University, Canberra, ACT, Australia
[d] NASA Goddard Institute for Space Studies and Center for Climate Systems Research, Columbia University

*Corresponding author:* Stephanie Blake (stephanieblake79@gmail.com)

**Abstract.** Explosive volcanism is an important natural climate forcing, impacting global surface temperatures and regional precipitation. Although previous studies have investigated aspects of the impact of tropical volcanism on various ocean-atmosphere systems and regional climate regimes, volcanic eruptions remain a poorly understood climate forcing and climatic responses are not well constrained. In this study, volcanic eruptions are explored in particular reference to Australian precipitation, and both the Indian Ocean Dipole (IOD) and El Nino-Southern Oscillation (ENSO). Using nine realisations of the Last Millennium (LM) with different time-evolving forcing combinations, from the NASA GISS ModelE2-R, the impact of the 6 largest tropical volcanic eruptions of this period are investigated. Overall, we find that volcanic aerosol forcing increased the likelihood of El Nino and positive IOD conditions for up to four years following an eruption, and resulted in positive precipitation anomalies over northwest (NW) and southeast (SE) Australia. Larger atmospheric sulfate loading during larger volcanic eruptions coincided with more persistent positive IOD and El Nino conditions, enhanced positive precipitation anomalies over NW Australia, and dampened precipitation anomalies over SE Australia.

## 1. Introduction

Volcanic eruptions have significant impacts on weather and climate variability through the injection of volcanogenic material into the atmosphere. Sulfate aerosols, formed through the reaction of $SO_2$ and $OH^-$ in the volcanic cloud, decrease incoming shortwave radiation, and if injected into the stratosphere, can generate a global response (Driscoll et al., 2012; LeGrande et al., 2016). Previous studies have identified relationships between volcanism and surface and tropospheric cooling (Driscoll et al., 2012), local stratospheric warming (Wielicki et al., 2002), strengthening of the Arctic Oscillation and Atlantic meridional overturning circulation (Oman et al., 2005; Stenchikov et al., 2006, 2009 & Shindell et al., 2004), and negative global precipitation anomalies (Gillet et al., 2004, Iles et al., 2013). The present study focuses on the under-studied relationship between large, globally significant tropical eruptions in the Last Millennium (850-1850CE) and Australian precipitation through examination of the direct radiative aerosol effect and the feedbacks of two tropical modes that strongly influence Australian rainfall: the El-Nino Southern Oscillation (ENSO) and Indian Ocean Dipole (IOD).

ENSO's effect on Australian precipitation has long been recognized. El Nino events typically cause averaged precipitation deficits, while La Nina cause positive precipitation anomalies (Meyers et al., 2007; Pepler et al., 2014). In addition, a statistical relationship has been demonstrated between explosive tropical volcanism and ENSO where large tropical eruptions can increase the likelihood and amplitude of an El Nino event in following years, followed by a weaker La Nina state (Adams et al., 2003). Further work by Mann et al. (2005), Emile-Geay et al. (2008), McGregor et

al. (2010), Wahl et al. (2014) and Predybaylo et al. (2017) supported this result. Pausata et al. (2015) and Stevensen et al. (2016) identified that a radiative forcing threshold value of more than 15 W m$^{-2}$ is required to affect the ENSO, and that high latitude Northern Hemisphere eruptions, in addition to tropical eruptions, are capable of doing so, as long as the forcing is asymmetric with regards to the equator.

The relationship between volcanic forcing and ENSO has been attributed to two contrasting, though not unrelated, mechanisms. The dynamical thermostat mechanism (Clement et al., 1996), whereby a uniform reduction of the surface heat flux due to volcanism causes warming of the eastern equatorial Pacific, was identified as the driver of ENSO's response to volcanism by Mann et al. (2005) and Emile-Geay et al. (2008). Conversely, a shift in the Intertropical Convergence Zone (ITCZ) induced by strong radiative forcing, was accredited in more recent studies (Pausata et al.,
2015; Stevenson et al., 2016). Preconditioning does impact the severity of the ENSO response. Predybaylo et al. (2017) found that years with an initial central Pacific El Nino ENSO phase show the largest statistical impact from Pinatubo-sized eruptions and that summer eruptions coincided with a more pronounced El Nino response.

Despite the understanding that volcanism can trigger or amplify El Nino events in the following years, the exact
relationship between ENSO and volcanic forcing is still debated. McGregor and Timmermann (2011) and Zanchettin et al. (2012) reported an enhanced probability of La Nina events occurring in the immediate years after a volcanic eruption, rather than El Nino, while several other studies (Self et al., 1997; Robock, 2000; Ding et al., 2014) found no relationship between ENSO and volcanic forcing. Robock (2000) argued that both El Chichon and Pinatubo reached their peak forcing after the initiation of El Nino events, indicating a coincidental relationship, while other studies
(Driscoll et al., 2012; Lewis & Karoly, 2014; Lewis & LeGrande, 2015; Predybaylo et al., 2017) have pointed out challenges in determining long-term characteristics of ENSO due to short instrumental records, and its relationship to volcanic forcing due to variable representations of both ENSO and volcanic aerosols in GCMs (Global Climate Models).

Comparatively little research has gone into the effects of volcanic forcing on the Indian Ocean Dipole (IOD), despite its known climatic impacts on Indian Ocean basin countries, such as Australia, South Africa, India and Indonesia (Cheung & Abram, 2016). The IOD is the zonal sea surface temperature (SST) gradient between the tropical western Indian Ocean (WIO) and the tropical south eastern Indian Ocean (EIO) (Roxy et al., 2011), defined by the Dipole Mode Index (DMI). Positive IOD (pIOD) states typically cause averaged precipitation deficits over Australia, and negative IOD
(nIOD) cause a surplus (Meyers et al., 2007; Pepler et al., 2014). While ENSO is often considered primarily responsible for triggering Australian droughts, the IOD has been shown to have an equal, if not larger, impact on heavily populated areas of Australia, with all significant southeastern Australian droughts in the 20$^{th}$ C showing a larger response to pIOD events than El Ninos (Ummenhofer et al., 2009).

Cheung & Abram (2016) found that the DMI shows a statistically significant correlation to volcanic forcing, with a negative IOD (nIOD) occurring immediately after an eruption and a positive IOD (pIOD) one year later. Maher et al. (2015) found a similar relationship, with coinciding El Nino and pIOD events occurring 6-12 months after the peak of volcanic forcing. The response of the IOD to volcanic forcing has been hypothesised to result from either the IOD's relationship with ENSO (Cheung & Abram, 2016), or the volcanically-induced reduction of the Asian Monsoon
(Anchukaitis et al., 2010; Zambri et al., 2017).

The direct radiative effect of volcanic aerosols has been found to cause global precipitation deficits for up to 5 years post-eruption (Robock & Lui, 1994; Iles et al., 2013; Gillett et al., 2004; Gu & Adler, 2011; Soden et al., 2002; Joseph & Zeng, 2011; Schneider et al., 2009; Timmreck et al., 2012; Iles et al., 2015). However, these deficits have been shown to vary seasonally (Joseph & Zeng, 2011), and cause positive precipitation anomalies over the NW and SE of Australia in the Southern Hemisphere (SH) winter and early spring (July to September; JASON), despite significant precipitation deficits in the summer (Joseph & Zeng, 2011; Schneider et al., 2009). This current study explores these relationships between volcanic eruptions and Australian rainfall during JASON, with reference to the direct radiative effect and the feedback effects of the ENSO and IOD.

## 2. Data and methods

### 2.1 Simulations

To understand the response of the IOD, ENSO and Australian precipitation to volcanic forcing in the Last Millennium, we analysed 9 ensembles from the NASA GISS ModelE2-R (hereafter simply GISS) (Schmidt et al., 2014). The GISS ensemble was run for the pre-industrial part of the Last Millennium (LM), from 850-1850 CE, which is defined by the Coupled Model Intercomparison Project Phase 5 (CMIP5) (Taylor et al., 2012). GISS is run at 2 degrees x 2.5 degrees horizontal resolution, with 40 vertical levels up to 0.1 hPa. The "non-interactive" atmospheric composition model, or NINT, is coupled with the dynamic Russell ocean model (Schmidt et al., 2014).

Evaluations of the accuracy of ENSO, IOD, surface temperature, precipitation and volcanic aerosols modelled in GISS have been conducted by Flato et al. (2013), Schmidt et al. (2014), Moise et al. (2015) and Miller et al. (2014). Global temperature observed at surface, middle troposphere and lower stratosphere are all accurate to within 2 standard deviations in the Historical ensemble (Miller et al., 2014), with GISS surface temperatures agreeing with observations in all areas of interest to this study to within 2°C and correctly simulating surface cooling following volcanic eruptions (Flato et al., 2013; Miller et al., 2014). In the Southern Ocean some systematic deficiencies cause large SST biases, however overall biases remain below 0.7-0.8°C (Schmidt et al., 2014).

Mean global precipitation is too high during the historical period, particularly around the tropics, when compared to observations (Schmidt et al., 2014), however the spatial pattern of precipitation agreed with trends calculated from Global Precipitation Climatology Project retrievals (Miller et al., 2014). Australian precipitation had a spatial-temporal root square mean error (RSME) of ~1mm/day and the model was deemed to provide good representations of surface temperature and precipitation over the entirety of Australia (Moise et al., 2015).

GISS captures the basic east-west structure of the tropical Pacific well, and follows the trend of the NINO3.4 index with the greatest accuracy between 2-3 years of an ENSO event (Flato et al., 2013; Schmidt et al., 2014). Calculation of the NINO3 index (150-90W and 6N-6S) for GISS and the Reynolds OI SST (https://www.ncdc.noaa.gov/oisst) observations for the first 50 years of the GISS piControl displayed an underestimation of ENSO intensity in GISS (Fig 1 (a,b)), meaning the modelled SST anomalies were about half as large as those observed. However, variability throughout the year is consistent with observations (Fig 1 (c,d)), and regression of the index onto SST, shows the global spatial pattern is also in good agreement, including the positive IOD index typically observed during ENSO events (Fig 1 (e, f)). Similar regression of the NINO3 index against precipitation (Fig 1 (g, h)) also showed spatial concurrence

between GISS and GPCP (https://precip.gsfc.nasa.gov/) observations, displaying the precipitation dipole over the Indian Ocean associated with the IOD.

Five ensembles were forced with volcanic forcing, while four were not to compare the effect of volcanic aerosols with non-volcanically influenced scenarios. Of the five run with volcanic forcing, four were forced with Crowley and Unterman (2013)'s aerosol optical depth data (CR), and one with double the Ice-core Volcanic Index 2 by Gao et al. (2008) (2xG) (see Table 1 for experiment summary). The LM was chosen for analysis as the period contains the majority of large tropical volcanic eruptions recorded in these datasets. The GISS model is forced with prescribed Aerosol Optical Depth (AOD) from 15-35 km, with a 4-layer vertical (15-20km, 20-25km, 25-30km and 30-35km and 24 layer (8°) longitudinally independent latitude, with Reff specified as per Sato et al. (1993). The LM simulations also include transient solar and land use histories that differ between ensembles. However, as this analysis focuses primarily on the immediate post-volcanic response, the impact of these smaller amplitude and slow-varying forcings is likely to be insignificant (Colose et al., 2016).

## 2.2 Methods

First, the six largest tropical eruptions between 850-1850CE were identified by the magnitude of their total global stratospheric sulfate aerosol injection (Tg) from the IVI2 Version 2 dataset, revised in 2012 (Gao et al., 2008), and the years surrounding eruption extracted for analysis (see Table 2 and Figure 2). Eruptions were deemed as tropical if volcanic aerosols were present in significant amounts in both hemispheres. The Kuwae eruption is included within the analysed eruptions, and is dated to 1452CE. While this year contains the bi-hemispheric deposition from the Kuwae eruption in both volcanic datasets used in this study, it is important to note that Sigl et al. (2013) recently constructed an ice-core record of volcanism that dates the Kuwae eruption to 1458/1459CE, with another, smaller eruption occurring at 1452. For the purposes of this paper, however, Kuwae will be considered as the 1452 deposition event.

We explored anomalous conditions in ENSO, the IOD and Australian rainfall. For ENSO, the period December-February (DJF) was examined using the NINO3.4 index (Trenberth, 1997), defined by the averaged sea surface temperature (SST) anomalies between 5N-5S and 170-120W. When analysing the IOD, the July-November (JASON) period was examined due to the tendency for pIODs to develop and mature over these months (Weller et al., 2014). The IOD was measured using the Dipole Mode Index (DMI), which subtracts the averaged SST in the EIO (90-110E; 10S-0) from the averaged SST in the WIO (50-70E; 10S-10N).

Australian precipitation was processed to find the anomalies of each season and year relative to the long-term mean over the JASON period. Analyses were conducted on area-averaged rainfall for the south-eastern Australian (132.5-155E; 27.5-45S) and north-western Australian (110-132.5E; 10-27.5S) regions. The south-east and north-west were chosen for analysis as the effect of the IOD on Australian precipitation is largest in, and potentially limited to, these general areas (Ashok et al., 2003).

The response of these large-scale modes of variability and rainfall are investigated using an epoch approach. For each major identified eruption, a response was defined by subtracting a reference period (the mean of 5 years pre-eruption) from the eruption year and the six years following eruption individually. A reference period of five years was chosen as it minimised the effect of trends or low-frequency climate variability (Iles et al., 2013). Mode specific graphs (IOD,

ENSO, Australian Precipitation) focused on the nine years surrounding eruption (years -2 to 6, with year 0 being the year of eruption). The mean of all six eruptions in each ensemble were calculated for individual years to reduce the noise associated with single eruptions, and then the mean of all ensembles included in each forcing category were compared (CR, 2xG or None).

The significance of the climatic response was assessed by comparing volcanically forced ensembles to the 'None' ensemble. The post eruption 'None' ensemble anomalies represent random sampling and have no physical relation to any eruption, and thus, their annual averages can be compared with those of the volcanically forced runs. One test of significance is shown by the 0.6 standard deviation threshold which represents the magnitude exceeded by natural variations only 5% of the time, identified by the two-variable t-test threshold for the CR runs. The threshold for the 2xG runs is slightly higher (0.9 standard deviation) due to the smaller number of forced simulations.

One ambiguity with this statistical test is that the anomaly of the 'None' ensemble doesn't directly correspond in time to the anomaly of the volcanic simulations. As an alternative test that uses only the volcanic ensemble, we calculated the likelihood that unforced variability could result in an anomaly relative to the five-year average prior to eruption (Ruxton, 2006). An anomaly magnitude exceeding 0.49 standard deviations results from unforced variability in the CR simulations only 5% of the time, indicating that our applied threshold is conservative. More generally, these statistical tests should be viewed as a guide rather than an intrinsic boundary for distinguishing post-eruption anomalies from unforced variations because of the subjectively chosen threshold for significant (here, 5%) and assumptions inherent to each test.

## 3. Results

The global SST response for the CR forcing group shows predominantly surface cooling anomalies (Fig. 3). More specifically, cooling occurs in the Northern Hemisphere from years 0-3, while the south Atlantic, Indian and Pacific oceans show mostly minor warming. The uniform reduction and re-distribution of surface temperature causes an El Nino-like warming temperature gradient in years 0, 1 and 4 post eruption. In the Southern Hemisphere, cooling is most pronounced over land masses, particularly Australia and the southern tip of Africa.

The DMI response showed a significant pIOD condition one year after major eruptions in all volcanically forced ensembles that persists from year +1 to +4, followed by an abrupt negative IOD (nIOD) in year +5 (Fig. 4). This response can also be seen in Fig. 3, where the EIO region shows larger and more widespread cooling anomalies than the WIO region in years 1, 2 and 4. This response contrasts to the non-volcanically forced ensembles, which show neither a prolonged pIOD nor nIOD condition.

The response of the DMI to the largest and smallest eruptions were also extracted. Fig. 5 show the mean DMI response to the 1258 Samalas eruption (257.91 Tg) and the 1600 Huaynaptina eruption (56.59 Tg). Our results show that while both eruptions caused a significant simulated pIOD at year 1, the larger 1258 Samalas eruption persisted with a significant pIOD condition in years 2 and 4, while the 1600 Huaynaptina eruption did not.

The mean NINO3.4 multi-volcano response to ensemble forcing showed a statistically significant El-Nino like response for all 6 years following eruption, with a peak at year 3 in both the CR and 2xG ensembles (Fig. 6). The non-

volcanically forced ensemble group showed neither a significant El Nino nor La Nina tendency, with the NINO3.4 index remaining within 0.4/-0.4. The index also showed an increase in the intensity and endurance, of post-volcanic El Ninos between the Samalas and Huaynaptina eruptions (Fig. 7). The Samalas eruption was followed by an El Nino that endured for 3 years, from years 1-3, peaking at a NINO3.4 anomaly of 0.68 in year 3, while Huaynaptina peaked at 0.53 in year 2 from an El Nino that endured for 2 years.

Fig. 8 shows the mean precipitation response of all volcanically forced ensembles. Precipitation deficits can be seen in the tropics in years 0-2, the most substantial and widespread of which occur in the Asian monsoonal area and the western Pacific basin. Bands of decreased precipitation also occur at approximately 40°S, between 0-40°N in the North Atlantic Ocean and between 40-80°N over Northern America and Europe. The Southern Hemisphere subtropics appear to have a slight increase in precipitation in years 0-2, most prominently over Australia and southernmost Africa, while the southern polar region (60-90°S) shows only variable minor precipitation anomalies occurring in all 6 years post eruption.

Ensembles with volcanic forcing showed an increase in precipitation over southeast (SE) (Fig. 11) and northwestern (NW) (Fig. 9) Australia between July to November (JASON). Both areas showed predominantly positive anomalies in years 0-5 post-eruption, with the largest response seen between years 0-2. SE Australia also shows large positive precipitation anomalies in year -1 resulting from unforced variability. NW Australia (Fig. 9) showed larger positive precipitation anomalies between years 0-2 than SE Australia (Fig. 11) in the CR ensemble mean, and in years 0 and 2 in the 2xG ensemble mean.

Comparison of the precipitation anomalies following the Samalas and Huaynaptina eruptions in NW Australia (Fig. 10) showed that the smaller eruption had a delayed and smaller positive precipitation peak, with Samalas peaking in year 0 with an anomaly of 0.23 and Huaynaptina in year 2 at 0.14. While the Huaynaptina eruption also showed a delayed peak in precipitation in SE Australia (Fig 12), the persistence of positive precipitation anomalies exceeded those of the Samalas eruption. Huaynaptina recorded values > 0.17 in years 1-2 and a value of 0.12 in year 4, all of which were larger anomalies than the peak of the Samalas eruption at 0.11 in year 1 (Fig. 12).

Figures 13 and 14 show multi-volcano mean anomalous changes to the surface wind direction (Fig. 13) and speed (Fig. 14) over the 5 years following eruptions in the CR forcing group. The most notable changes occur in years 0 and +1 where anomalously strong Southern Hemisphere westerlies and anomalously weak south easterly trade winds occurred, accompanied by strong north-easterlies off the south-east coast of China and an intensification of North Atlantic circulation. In year +3 anomalous south easterly winds off the south-western coast, and anomalous westerlies off the central western coast, of South America are seen.

**4. Discussion and conclusions**

Our results suggest that the large-scale IOD and ENSO systems, and Australian rainfall regimes, were all impacted by large tropical eruptions of the Last Millennium.

The DMI response simulated in the GISS ensemble following large eruptions is complimentary to previous research conducted by Cheung & Abram (2015) and Maher et al. (2015). The pIOD peak in year 1 (Fig. 4) is consistent with

both studies, in which statistically significant pIOD conditions occurred between 6 months to 2 years after an eruption. Cheung & Abram (2015) also found a statistically significant negative condition immediately after eruption at year 0, however this was absent from both Maher et al.'s (2015) results and the CR forcing category in this study. The 2xG category does show a nIOD condition at year 0, but is not believed to be a response to volcanic forcing as a similar nIOD condition can be seen at year -1. The abrupt shift to a negative condition at year 5 was not found in either Cheung & Abram (2015) or Maher et al. (2015)'s results. Both studies found a gradual decrease in DMI from year 1 to years 3-4.

The smooth transition to a lower DMI following eruptions found by Cheung & Abram (2015) and Maher et al. (2015) contrasts with the abrupt change from a pIOD of approximately 0.13 at year 4, to an nIOD of -0.069 in the CR ensembles and -0.083 in the 2xG ensemble at year 5 (Fig. 4). This inconsistency between studies could be due to the selection of eruptions analysed by each paper. Cheung & Abram (2015) included all eruptions from 850-2005CE recorded on the IVI2 in their analysis. While this encompasses all eruptions analysed here, it also included many smaller eruptions that would likely have dampened the climatic response, a response that has been analysed in previous papers (Zambri & Robock, 2016). Maher et al. (2015) looked at the five largest eruptions from 1880 to present, of which the largest was Pinatubo (1991), measured at 30.10 Tg globally on the IVI2 (Gao et al., 2008). In comparison, our research deals with eruptions of much larger atmospheric loading, ranging from 56.59 to 257.91 Tg.

Therefore, the persistence of a high pIOD through to year 5 seen here may result from the larger mean atmospheric sulfate loading imposed. This theory is supported by the comparison between the Samalas and Huaynaptina (Fig. 5) eruptions. Our results showed that while both eruptions caused a significant pIOD at year 1, the larger 1258 Samalas eruption alone persisted with a significant pIOD condition in following years. Further support can be gathered from the comparison between the 2xG and CR ensemble means in fig. 4. Years 0-3 show more extreme values in the 2xG ensemble mean, while years 4-6 show similar values for both forcing categories. Maher et al. (2015) found a similar response, with the two largest eruptions analysed in the paper showing the largest and longest enduring pIOD anomalies. This suggests that larger mean atmospheric sulfate loading can cause not only more persistent, but also more extreme pIOD conditions.

The phase and intensity of the IOD is known to be influenced by the Asian monsoon (Brown et al., 2009; Xiang et al., 2011), and the physical mechanisms driving the pIOD response to volcanic forcing in GISS likely stems from this relationship. In GISS, the Asian monsoon was suppressed by the anomalous north easterly flow off the south-east coast of China in years 0 and +1 (Fig. 13; Fig. 14) generated by volcanic aerosols, and a decrease of convection over the warm pool, cause by El Nino-like anomalies in those same years (Fig 3). These feedbacks caused a comparatively warmer WIO, generating a pIOD. The Asian monsoon suppression following volcanic eruptions was also noted by Stevenson et al. (2016).

The NINO3.4 response found in this research supports previous studies by Adams et al. (2003), Mann et al. (2005), Emile-Geay et al. (2008), McGregor et al. (2010) and Maher et al. (2015), despite GISS modelling weaker SST anomalies than observations (Fig. 1 (a,b)). Fig. 6 shows a very prominent and persistent El-Nino response in all 6 years following eruption, however it lacks the weaker La Nina-like state that was observed 3-6 years after eruption in these previous papers. Spatial maps of SST (Fig. 3), while dominated by the overall volcanic cooling, show an El Nino-like pattern in the eastern Pacific that is most visible in year 4, possibly driven by the anomalous winds off the western coast

of South America in year +3 (Fig. 13). but is also distinctive in years 0, 1 and 3. We can therefore conclude that El Nino-like anomalies were generated in the multi-ensemble mean response in years 0-6 following eruption by a uniform reduction in surface temperature, driven by a decrease in the surface heat flux, a response also observed by Mann et al. (2005) and Emile-Geay et al. (2008), and that the intensification of the El Nino in year +4 was due to anomalous wind flow in year +3. Comparison of the Samalas and Huaynaptina (Fig. 7) eruptions also suggest that, similar to the DMI, the intensity and endurance of the ENSO response to volcanic forcing increases with increasing mean atmospheric sulfate loading. This once again supports the findings of Maher et al. (2015) that identified a similar pattern.

The positive response of Australian precipitation to volcanic forcing as seen here (Fig. 8, 9 &11) is in agreement with several papers that identified a precipitation surplus over Australia following large volcanic eruptions (Schneider et al., 2009; Joseph & Zeng, 2011). Our results suggest that the direct effect of volcanic aerosols on precipitation overrode the impact of the IOD and ENSO on Australian precipitation in the years following large tropical volcanic eruptions. NW Australia (Fig. 9) showed larger positive precipitation anomalies between years 0-2 than SE Australia (Fig. 11) in the CR ensemble mean, and in years 0 and 2 in the 2xG ensemble mean. This could be due to the positive precipitation anomalies that can be generated by combined El Nino and pIOD events in the NW Australian region (Meyers et al., 2007 & Pepler et al., 2014), enhancing the precipitation surplus caused by volcanic aerosols.

The varying response of NW Australia to the Samalas and Huaynaptina eruptions (Fig. 10) also supports the enhancement of the volcanically induced precipitation surplus by combined El Nino and pIOD events. The Samalas eruption was followed by strong and enduring El Nino and pIOD conditions for up to 4 years post volcanism, and showed larger positive precipitation anomalies from years 0-3 than the Huaynaptina eruption, that was accompanied by smaller, shorter-lived El Nino and pIOD conditions. The precipitation surplus to the Samalas eruption in NW and SE Australia also peaked earlier than Huaynaptina, which could be a response to the larger atmospheric sulfate loading. Interestingly, previous papers have not reported a relationship between atmospheric sulfate loading and the peak in precipitation response (Robock & Lui, 1994; Iles et al., 2013; Iles et al., 2015).

The precipitation anomalies of SE Australia (Fig. 12) further supports this theory. The response to the Huaynaptina eruption, while peaking later than Samalas, endured longer, and with larger positive anomalies. The effect of strong, combined El Nino and pIOD conditions on SE Australia is significant precipitation deficits (Meyers et al., 2007 & Pepler et al., 2014), and could explain the negative precipitation anomalies that occur in the Samalas response from year 2 onwards, where the combined influence of a strong El Nino and pIOD dampened the positive precipitation response generated by the atmospheric sulfate loading. It should be noted that the surprisingly large positive precipitation anomaly at year -1 for SE Australia is an indicator of uncertainty of the post-eruption values, however the significance of this event can still be seen by the comparison of the volcanic ensembles to the 'None' ensemble.

We note that our study has provided an analysis of climatic response to a set of forcings in a single climate model, which may limit the precise interpretation of responses to eruptions. Overall, volcanic aerosols remain an understudied climatic forcing such that the timing, magnitude and spatial footprint of past eruptions remains uncertain (Colose et al., 2016). In addition to uncertainties around the fundamental physical forcings, limitations still exist in the implementation of volcanic eruptions in climate models (Colose et al., 2016; Zambri et al., 2017). For example, Colose et al. (2016) suggest that improvements in model representations of volcanic particle size may improve the accuracy of model simulations. Furthermore, LeGrande et al. (2016) note that the chemistry and composition of a volcanic plume affects

its climatic impact, which requires realistic representation in climatic models. Overall, these limitations in modeling eruptions and the idealised approach adopted here may mean that impacts simulated do not precisely match those of the proxy record.

In summary, this paper aimed to identify the impact of large, tropical volcanism on the ENSO, IOD and Australian rainfall by averaging over multiple major eruptions and multiple simulations to reduce the noise associated with analysis of single volcanic eruptions, as seen by the 90[th] and 10[th] percentiles in fig 5, 7, 10 and 12. Results showed an El Nino and pIOD response in the immediate years following eruption, accompanied by positive precipitation anomalies over SE and NW Australia. The positive precipitation anomalies suggest that volcanic aerosol cooling dominates the precipitation response, rather than the effect of ENSO or IOD, despite aerosols also proving to be an important influence on these large-scale modes. Although this study focused on Australian rainfall regimes and its main climatic drivers, this approach can be applied for exploring the impact of time-evolving forcings, such as volcanism, in other regions.

### Acknowledgements

We thank NASA GISS for institutional support. S.C.L is funded through Australian Research Council (ARC) DECRA Fellowship (DE160100092) and additional funding is provided through the Australian Research Council Centre of Excellence for Climate System Science (CE110001028). We also thank the NASA MAP programme for continued support of A.N.L and R.L.M. Resources supporting this work were provided by the NASA High-End Computing (HEC) Program through the NASA Center for Climate Simulation (NCCS) at Goddard Space Flight Center.

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

| Volcanic forcing | Ensembles |
|---|---|
| None | E4rhLMgTs, E4rhLMgTnck, E4rhLMgTKk, E4rhLMgTk |
| Crowley & Unterman (2012) | E4rhLMgTncck, E4rhLMgTKck, E4rhLMgTcs, E4rhLMgTck |
| 2 x Gao et al. (2008) | E4rhLMgTKgk |

**Table 1: Volcanic forcing used for ensembles.**

| Eruption Name | Year | Tg |
|---|---|---|
| Samalas | 1258 | 257.91 |
| Kuwae | 1452 | 137.50 |
| Tambora | 1815 | 109.72 |
| Unknown 1 | 1227 | 67.52 |
| Unknown 2 | 1275 | 63.72 |
| Huaynaputina | 1600 | 56.59 |

**Table 2: The six largest tropical eruptions of the Last Millennium and their total global stratospheric sulfate injection (Tg) as recorded by Gao et al. (2008), revised in 2012.**

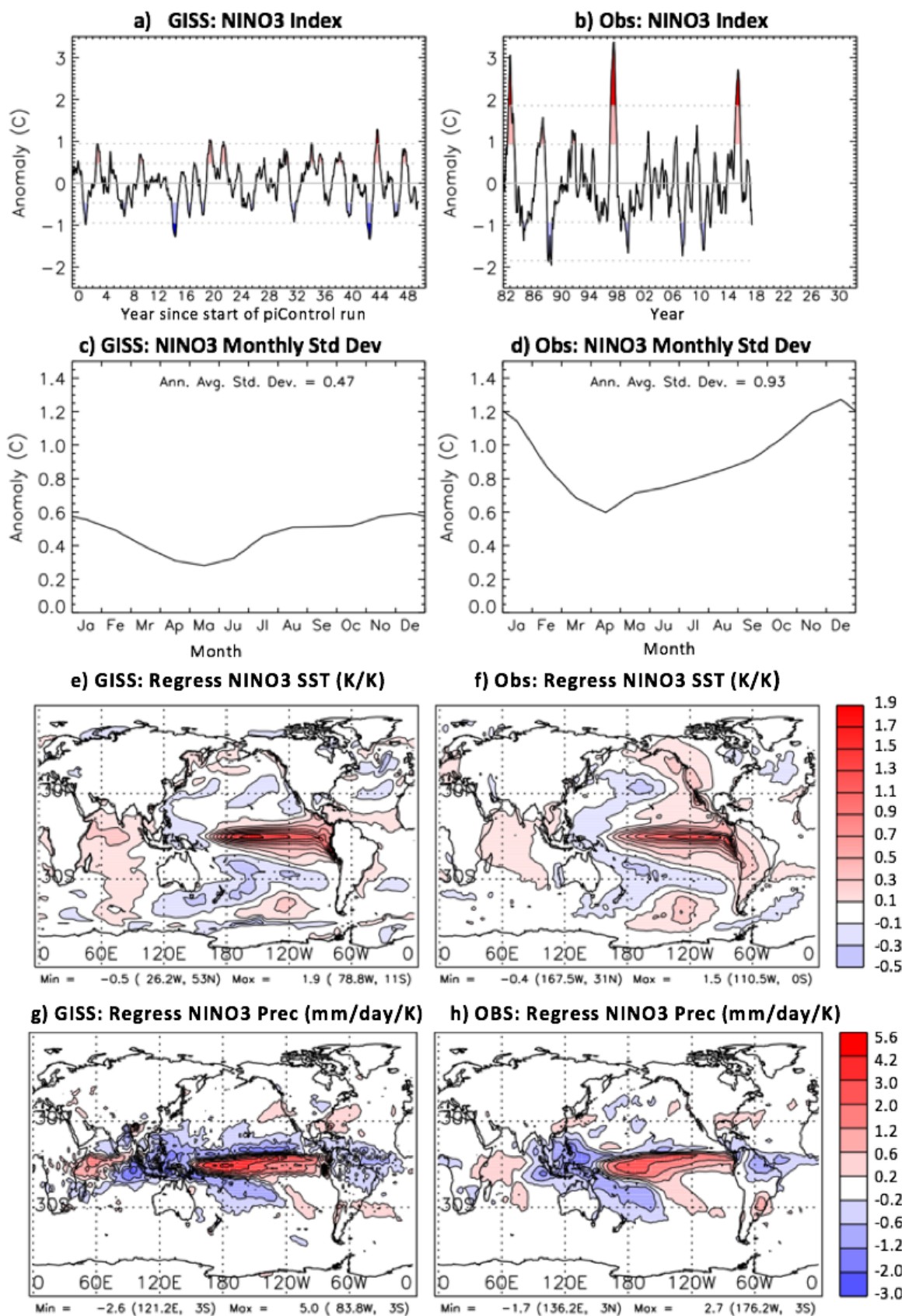


**Fig 1: Evaluation of GISS model ENSO variability against Reynold's OISST (1982-2017) and GPCP (1979-2017) observations. Model values are calculated using the first 50 years of the E2-R piControl simulation, where forcings are**

**absent. a, b)** NINO3 index created from a spatial average of GISS and Reynolds SST between 150W and 90W and 6S to 6N. The meridional extent used to construct the NINO3 index is typically 5S to 5N, but broadened here to correspond to archival grid for GISS SST. The index is constructed from monthly anomalies with no smoothing. **c, d)** Seasonality of ENSO variability, defined as the standard deviation of the NINO3 index for each month. **e, f)** The sensitivity of SST to NINO3.4 index derived by linear regression. **g, h)** The sensitivity to the NINO3.4 index derived by linear regression.

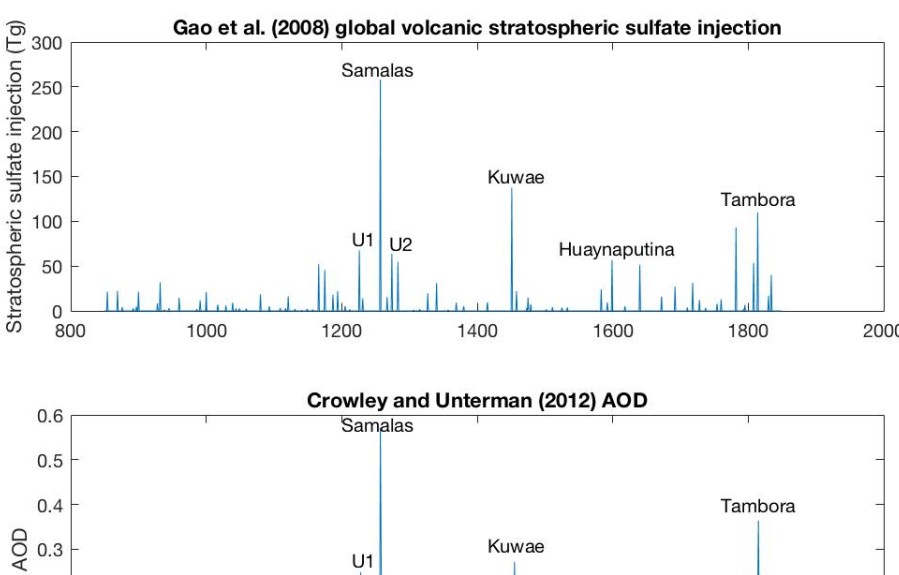

**Fig 2: Timeseries of volcanic forcing from Gao et al. (2008) (upper) and Crowley and Unterman (2012) (lower). The specific subset of volcanic eruptions investigated is labelled.**

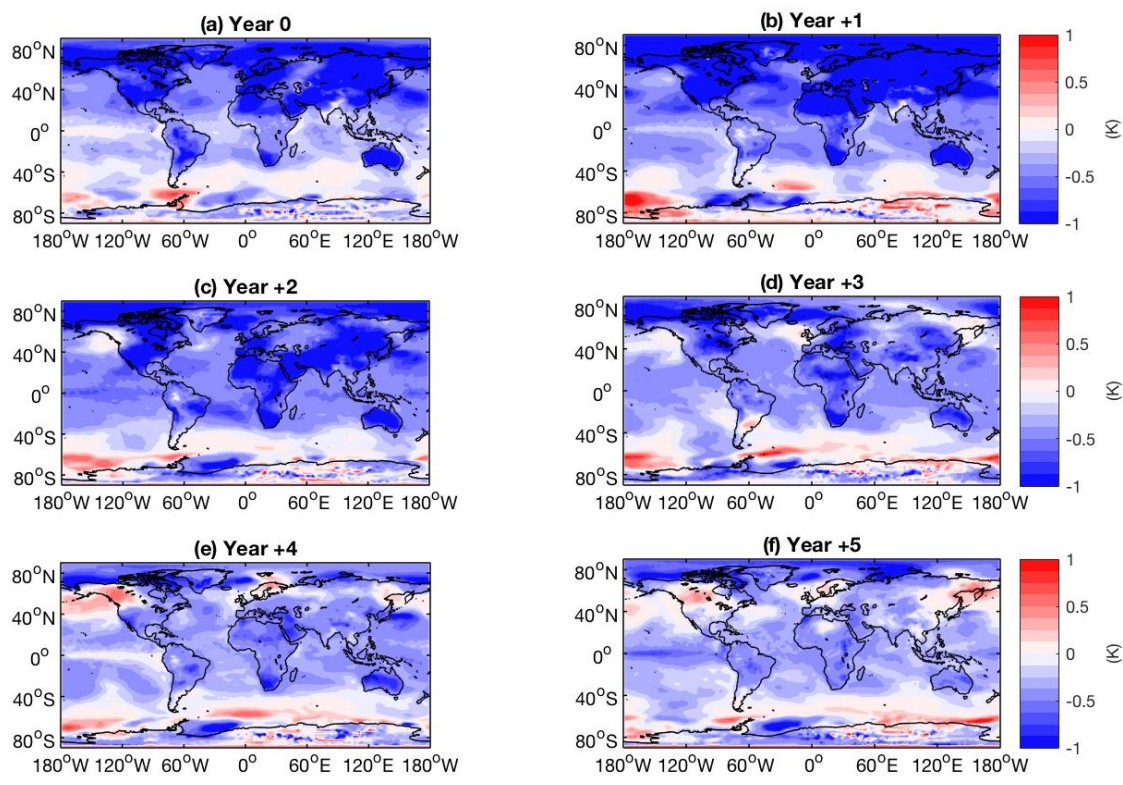

**Fig 3: Global SST anomalies (K) in response to the Crowley (CR) forcings, showing multi-ensemble mean response over JASON averaged across all analysed eruptions for years 0 to +5 after eruption**

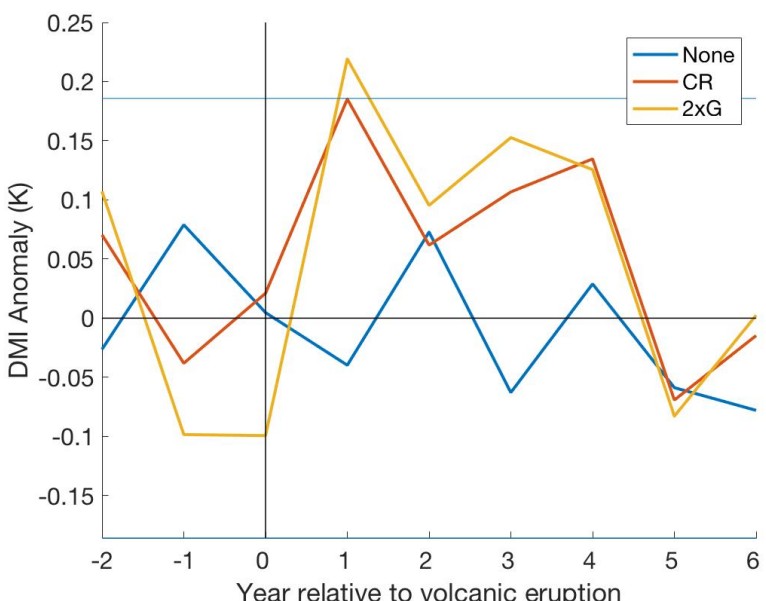


**Fig 4: Multi-ensemble and multi-volcano mean DMI (Dipole Mode Index) response to CR and None forcing groups and multi-volcano mean DMI response to the 2xG forcing group over July-November (JASON). Significance was tested using the 0.6 standard deviation threshold (horizontal blue lines), and by comparing the CR and 2xG ensembles to those without volcanic forcing.**


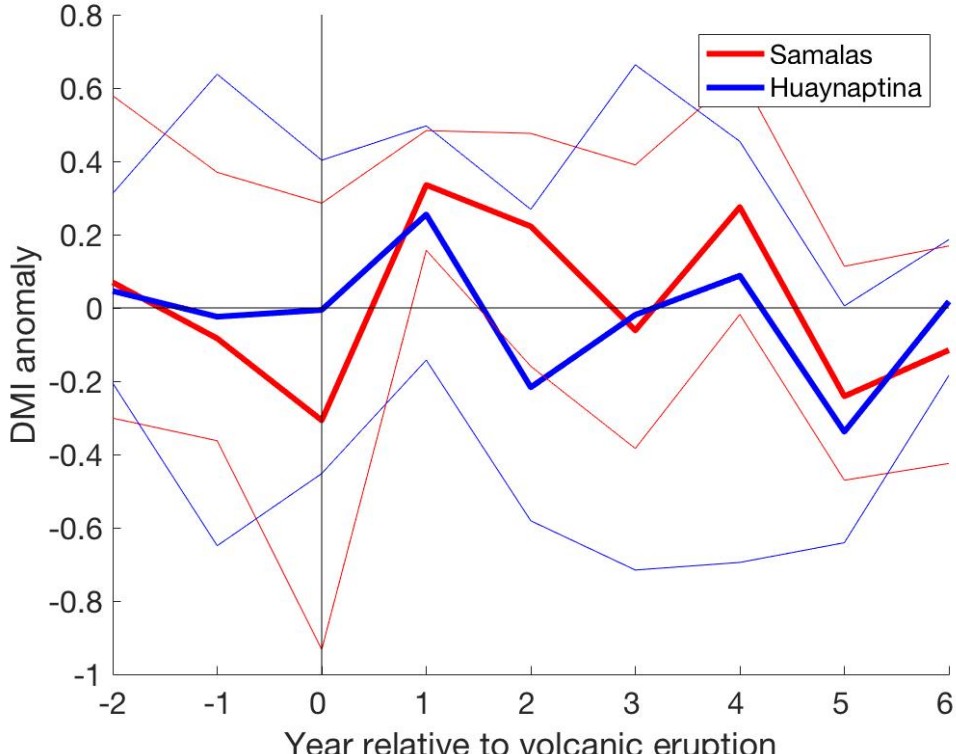

**Fig 5: Mean DMI response of all volcanic ensembles (four CR ensembles and one 2xG ensemble) to the largest (1258 Samalas) and smallest (1600 Huaynaputina) eruptions analysed over JASON. The bold lines represent the mean of all volcanic ensembles to each eruption, and the fainter lines represent the 90[th] and 10[th] percentile of the ensemble members.**

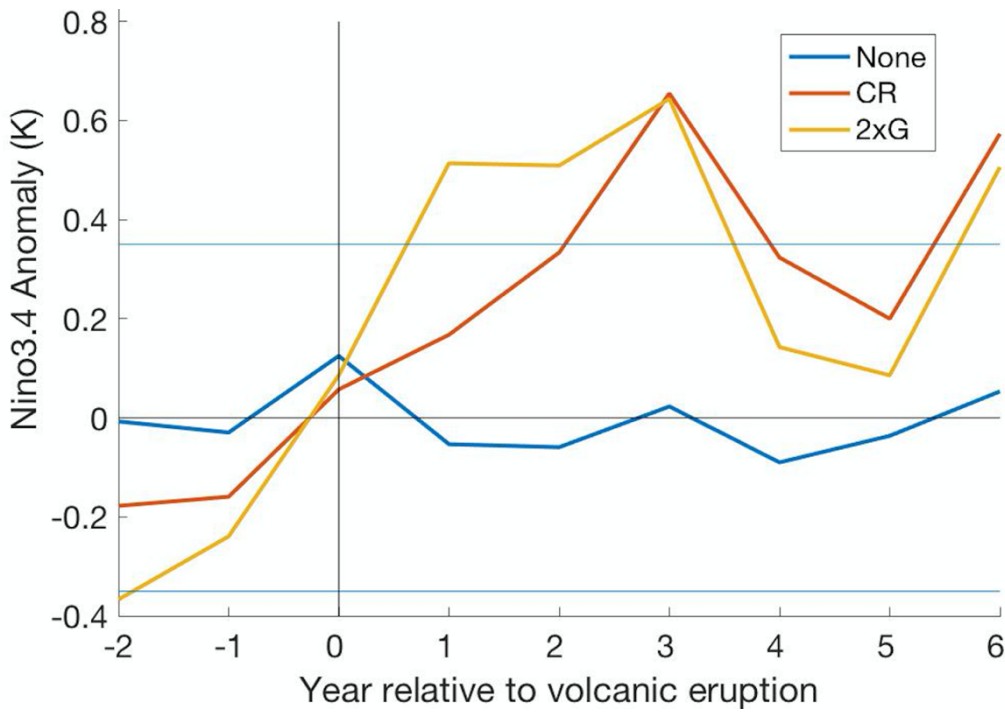

**Fig 6: Multi-ensemble and multi-volcano mean NINO3.4 response to CR and None ensemble forcing groups and multi-volcano mean DMI response to the 2xG forcing group over DJF. Significance was tested using the 0.6 standard deviation threshold (horizontal blue lines), and by comparing the CR and 2xG ensembles to those without volcanic forcing.**

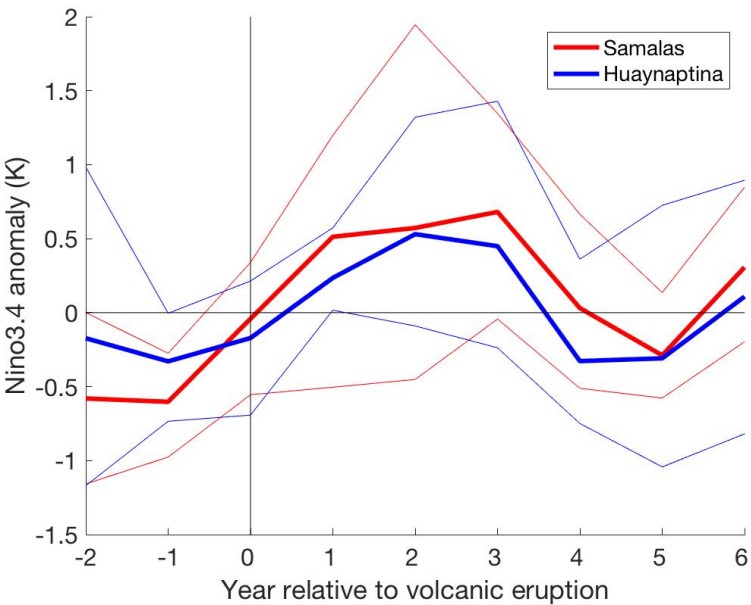

**Fig 7: Mean NINO3.4 response of all volcanic ensembles (four CR ensembles and one 2xG ensemble) to the largest (1258 Samalas) and smallest (1600 Huaynaputina) eruptions analyzed over DJF. The bold lines represent the mean of all volcanic ensembles to each eruption, and the fainter lines represent the 90[th] and 10[th] percentile of the ensemble members.**

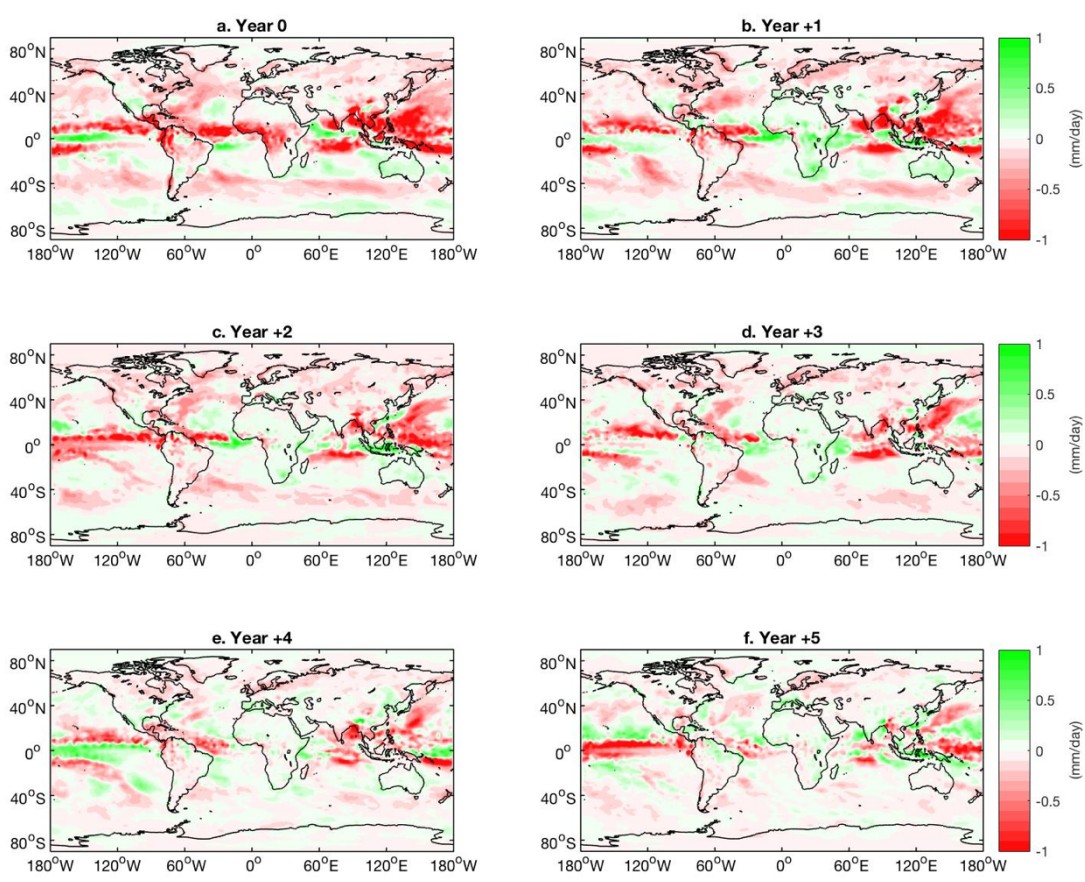

**Fig 8: Global precipitation anomalies (mm/day) in response to the Crowley and Unterman (CR) forcing, showing multi-ensemble mean responses over JASON averaged across all analysed eruptions for years 0 to +5 after eruption.**

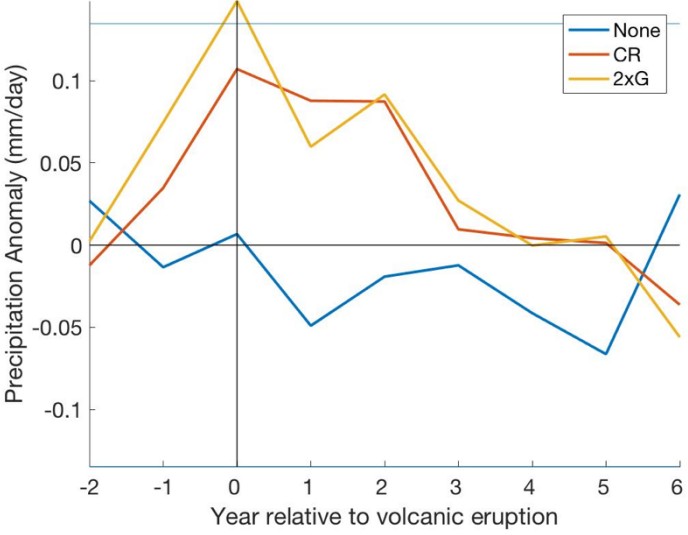


**Fig 9: Multi-ensemble and multi-volcano mean NW Australian precipitation (mm/day) response to CR and None forcing groups and multi-volcano mean to the 2xG forcing group in the eight years surrounding eruption over JASON. Significance was tested using the 0.6 standard deviation threshold (horizontal blue lines), and by comparing the CR and 2xG ensembles to those without volcanic forcing.**

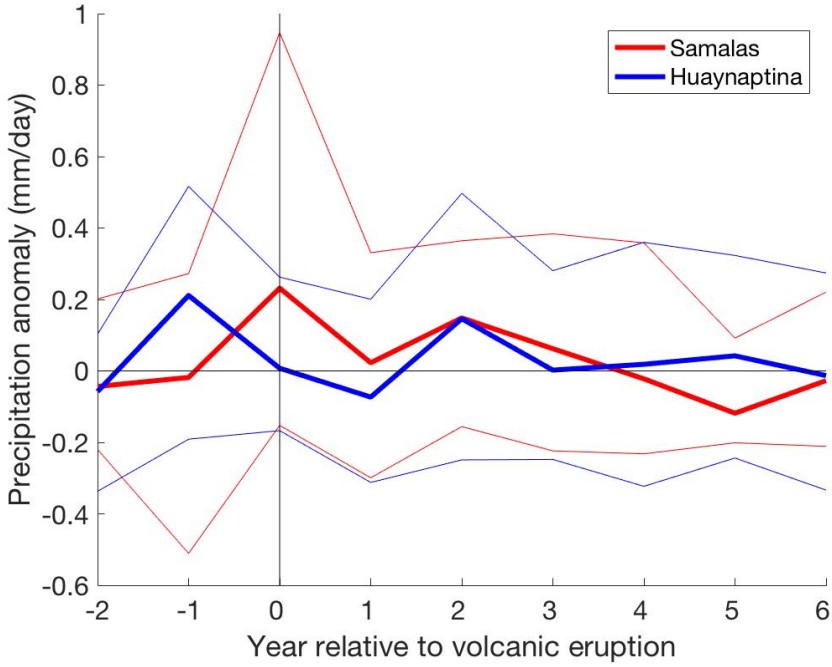


**Fig 10: Mean NW Australian precipitation (mm/day) response of all volcanic ensembles (four CR ensembles and one 2xG ensemble) to the largest (1258 Samalas) and smallest (1600 Huaynaputina) eruptions analysed over JASON. The bold lines represent the mean of all volcanic ensembles to each eruption, and the fainter lines represent the 90th and 10th percentile of the ensemble members.**

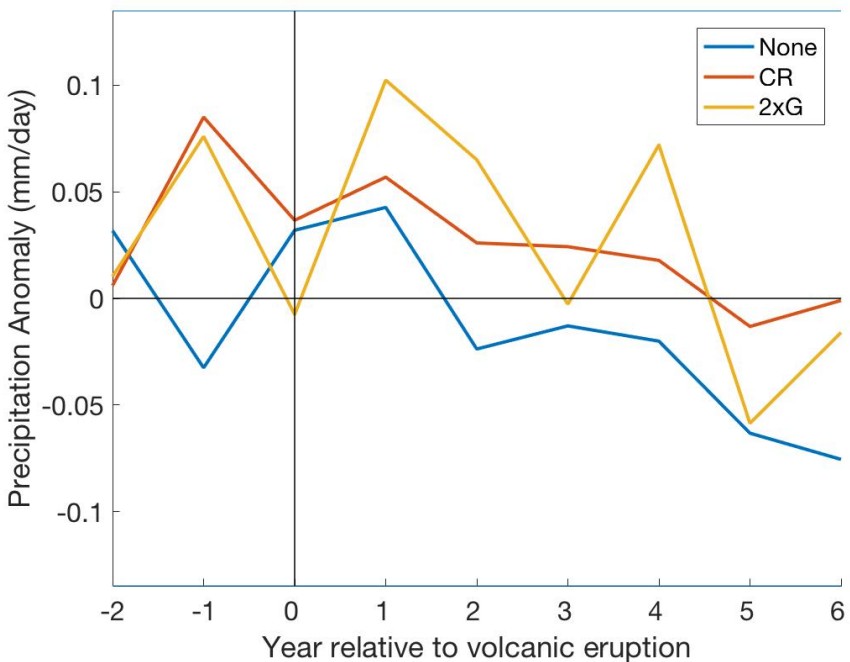


**Fig 11: Multi-ensemble and multi-volcano mean SE Australian precipitation (mm/day) response to CR and None forcing groups and multi-volcano mean to the 2xG forcing group in the eight years surrounding eruption over JASON. Significance was tested using the 0.6 standard deviation threshold (horizontal blue lines), and by comparing the CR and 2xG ensembles to**

**those without volcanic forcing.**

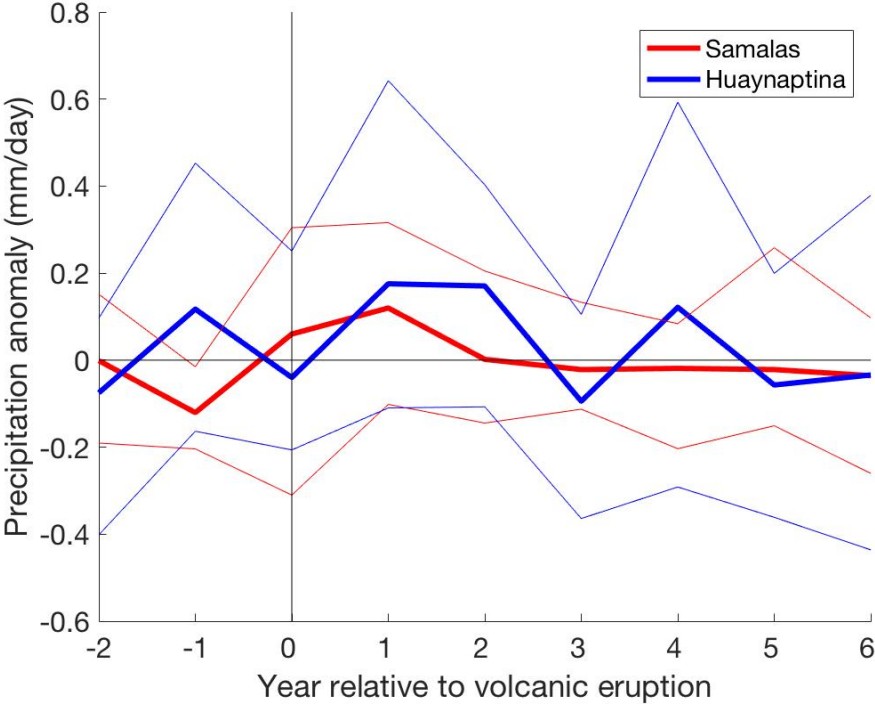

**Fig 12: Mean SE Australian precipitation (mm/day) response of all volcanic ensembles (four CR ensembles and one 2xG**

**ensemble) to the largest (1258 Samalas) and smallest (1600 Huaynaputina) eruptions analysed over JASON. The bold lines represent the mean of all volcanic ensembles to each eruption, and the fainter lines represent the 90[th] and 10[th] percentile of the ensemble members.**


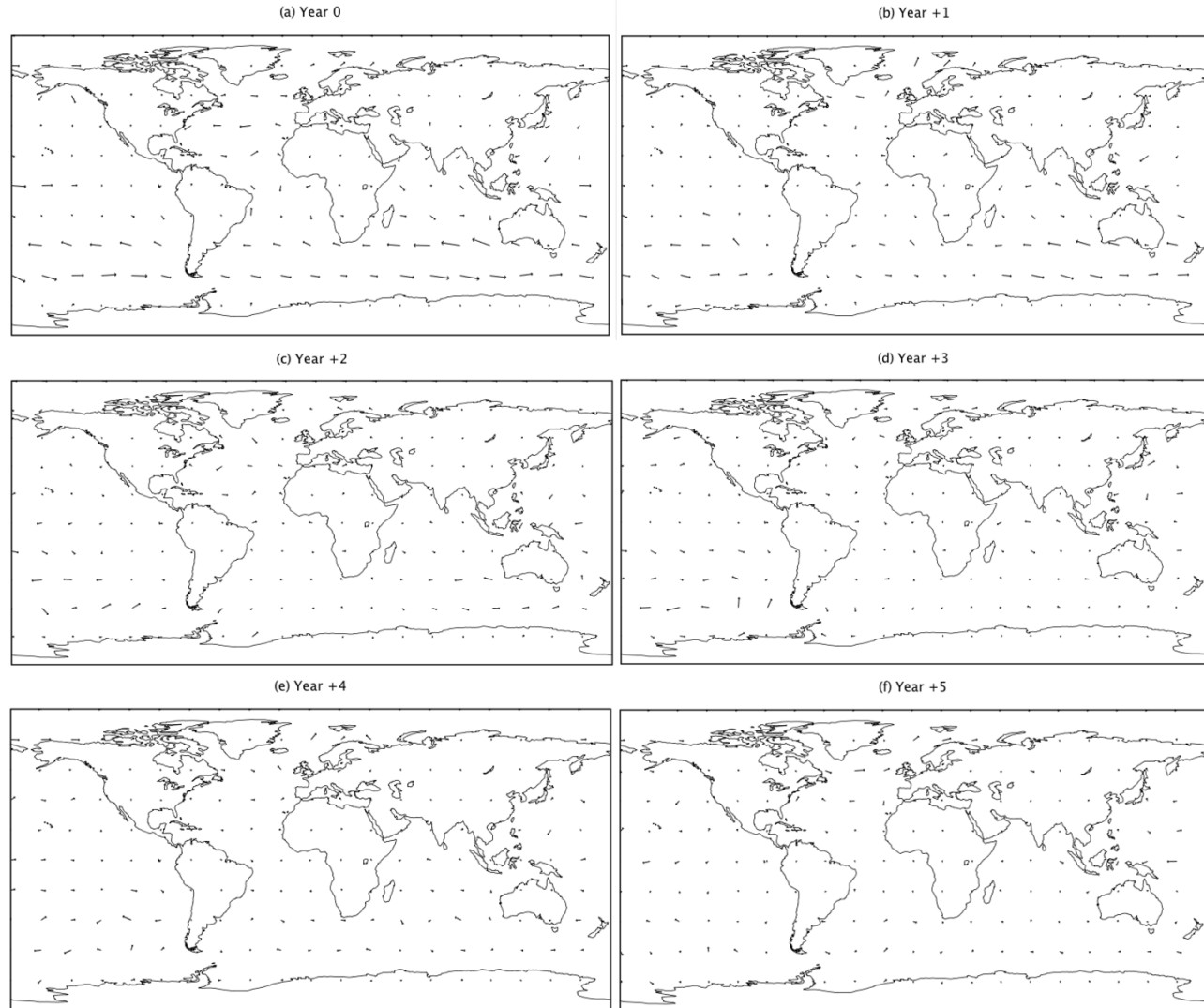

**Fig 13: Global surface wind direction anomalies in response to the Crowley and Unterman (CR) forcing, showing multi-ensemble mean responses over JASON averaged across all analysed eruptions for years 0 to +5 after eruption.**

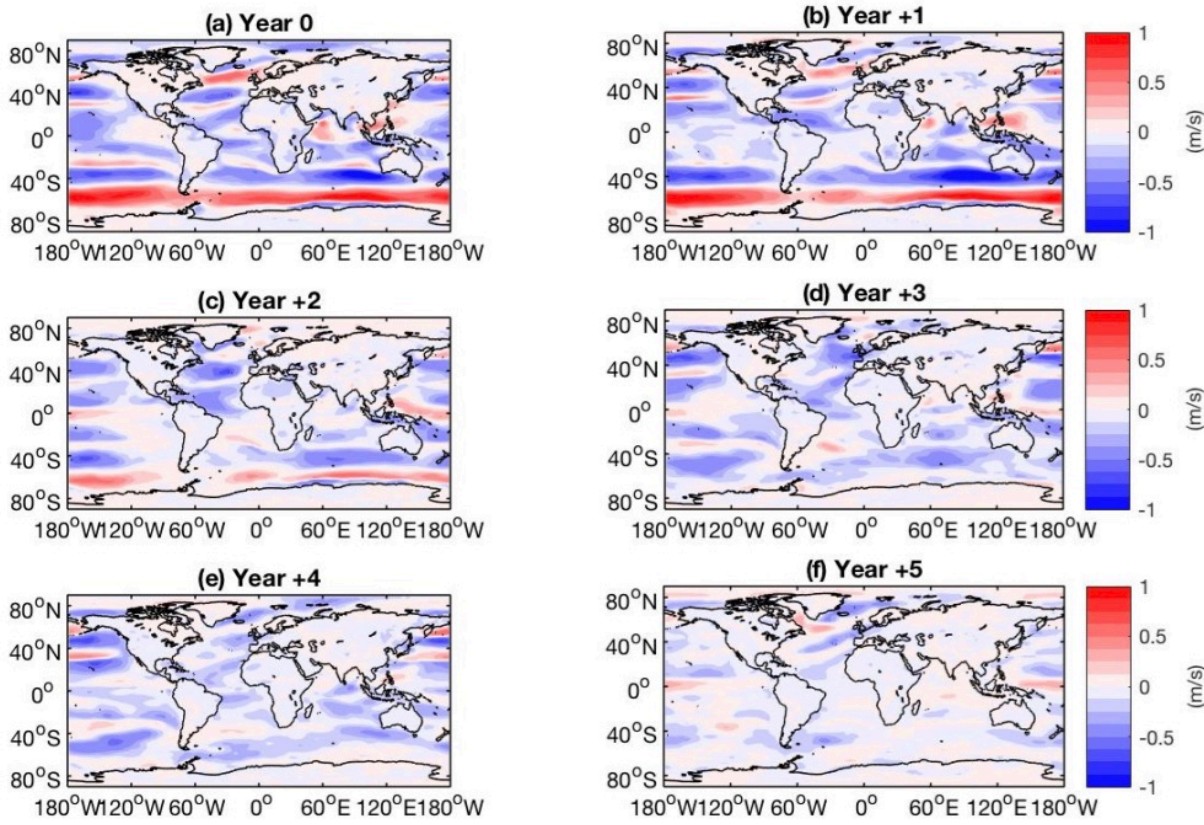

**Fig 14: Global surface wind speed (m/s) anomalies in response to the Crowley and Unterman (CR) forcing, showing multi-ensemble mean responses over JASON averaged across all analysed eruptions for years 0 to +5 after eruption.**