# Peer review of "Assessing the impact of large volcanic eruptions of the Last Millennium on Australian rainfall regimes"

_Climate of the Past, 2017_

## Referee Comment (RC1) · Anonymous Referee #1 · 3 Oct 2017

GENERAL REMARKS:

The manuscript "Assessing the impact of large volcanic eruptions of the Last Millennium on Australian rainfall regimes" in consideration for publication in Climate of the Past investigates the impacts of tropical explosive volcanic eruptions on Australian rainfall regimes, ENSO and the Indian Ocean Dipole. The authors use a set of specific global climate model realisations to address questions about the importance of volcanic forcing power and timing of the above mentioned impacts.

Generally, the topic of the manuscript fits Climate of the Past and is a valuable contribution to the community, The study sheds light on volcanic climate impacts in the

Southern Hemisphere, for which only few studies exist so far.

However, I feel like the manuscript lacks content in some important parts. In my opinion, the three main weaknesses are:

1) Experimental design: Since the study largely focuses on the IOD and ENSO, the authors should describe in detail how SSTs and evaporation over oceans is handled in the model. Is it an AMIP run? Is it coupled? Slab ocean? And if the SSTs are prescribed, can it be used at all? These things are completely disregarded in the manuscript at hand.

2) Model evaluation: There is NO effort made to convince me that this model is doing a good job in terms of precipitation. I would like to see a comparison between model and proxy data, or model and (PDSI) reconstructions.

3) Physical mechanisms behind the shown effect on ENSO and IOD: Although a lot of studies are cited as to show that there is agreement with previous work, I didn't find a proper physical explanation for the effects we see in the plots besides the paragraph in the introduction. I would like to see a discussion about the mechanisms in this specific model.

See below for specific remarks.

SPECIFIC REMARKS:

line 34: I would add here newer studies about precipitation like Iles et al 2013 or Wegmann et al 2014.

line 93: Not sure what exactly is unclear? Do you mean dynamic vs. radiative changes? Please explain further what gap your study is filling.

line 103: Here I really would like to see a thorough description of the ocean setting for this model (see above). What kind of SSTs are used? Is there a dynamic ocean? How is the volcanic signal supposed to show in the SSTs? Is it valid to use the model to

investigate ocean surface changes? CMIP5 also had AMIP runs included, so I can't really infer if your model was coupled or not.

line 103: I assume CO2 forcing is done in CMIP5 fashion?

line 148: As I said before, I would like to see an evaluation for starters. How good is the model in terms of precipitation? The comparison doesn't have to be for Australia, but I want to be convinced that the model gets the broader precipitation response. Otherwise, the rest of the results is less meaningful.

Figure 2: Is this the annual mean? If so please indicate that fact. Maybe it would be nice to see DJF and JJA anomalies? Is the model able to do the NH winter warming? And if not, is there an argument to make that the model doesn't get the dynamics right (as is the case with many CMIP5 models)?

line 176: Again, I wonder about NH winter warming. Should counteract the summer cooling over the NH.

line 194: Indeed it suggests that. But how did it work? Where is the discussion of the physical mechanism? How is the signal transported in the model? Maybe show evaporation, heat content and other metrics to show the mechanism in the model.

Line 241: Okay, but how does this effect override the impact of IOD?

Table 1: It says strongest eruptions in the last millennium but I am pretty sure that Tambora was stronger than Huaynaputina as Figure 1 shows. In fact even the unknown 1809 eruption is bigger. Please adjust your Table description.

Figure 3: in fact 2xCG is not an ensemble and should be marked as such.

Figure 4: The 90th percentile of what? The ensemble members? If so, please adjust. (also for the rest of the Samalas & Huaynaputina plots.

Figure 10: Here it says Australian precipitation whereas Figure 9 says NW Australian precipitation and Figure 11 says SE Australian precipitation. I assume Figure 10 also

is SE precipitation?
* * *

---

## Referee Comment (RC2) · Anonymous Referee #2 · 12 Oct 2017

This work analyzed the precipitation response to large volcanic perturbation in Australian, using GISS Model E millennium simulations. The topic is an important contribution to the systematic assessment of global hydroclimate response to the volcanic radiative influence. The paper is well written; the results are clearly presented and discussed. I would recommend a major revision of the paper by addressing the following issues, before it could accepted for publication:

1. In the "Introduction" section 1.1 It is not clear from the description which relationships between volcanic eruptions and Australian rainfall are unclear and are explored in this paper. Please be more specific. A significant part of the introduction is devoted to

the literature review of the volcanism and ENSO relationship, while the focus of this study is on volcanic eruptions and Australian rainfall. Please modify either the title, or the structure of introduction. For example, the authors may consider introduce the role of ENSO in changing Australian rainfall before discuss the volcanism and ENSO. relationship

2. In the "Data and Method" section 2.1 Line 99, please explain briefly why the last millennium period is especially chosen. 2.2 Please provide a short description of model performance, especially those closely related to the ENSO, IOD and volcanic climate responses. Please also provide a short explanation of why the five volcanically forced scenarios were chosen. 2.3 Line 134, please also discuss the ENSO impact in the chosen north-west and south-east regions.

3. In the "Results" section 3.1 Please explain the advantages of comparing the response between the largest 1257 Samalas and the smallest 1600 Huaynaptina eruption, rather than a series comparison among the 6 eruptions of various size. 3.2 Line 164, there is no ensemble runs for the $2{\times}GC$ case, please make the distinction . 3.3One of Figure 8 or Figure 9 should be "SE (instead of NW)Australia" response. Please also correct the reference to the Figure 10 in Line 179. The use of "multi-model mean" in several figures is misleading, please consider change to model ensemble. 3.4 The focus of this work is Australia, however, there is no spatial figures dedicated to the particular area of study. 3.5 Please verify the use of 0.6 standard deviation as the significant level.

4. In the "Discussions and Conclusions" section 4.1 Line 218-221, the difference between Samalas and Huaynaptina response does not seem significant and the use of it as support for the persistence of a high pIOD sounds weak to me. Please consider to use all six (or even more eruptions) of different sizes to analyze the role of eruption magnitude. 4.2 Line 228-235, I do not see the El Nino response persist from year 0 to year 6 in both Fig. 5 and Fig.2. Please explain why the El Nino-like pattern in the eastern Pacific is most visible in year 4, but not earlier. 4.3 Line 241-243, please

demonstrate in more detail how did the direct precipitation effect of volcanic aerosols override the impact of the IOD on Australian precipitation. Which parameter represents the direct precipitation effect of volcanic aerosols? How was the override effect appear in the results? By the time difference? 4.4 The most important advantage of using modeling results is the capability of exam the physical mechanisms behind the shown effects (such as the impact on ENSO and IOD, and their influence on Australian rainfall). Please provide some discussion of mechanism using the original results from this paper, rather than referring to previous studies.

Please also note the supplement to this comment:
https://www.clim-past-discuss.net/cp-2017-109/cp-2017-109-RC2-supplement.pdf
* * *

---

## Short Comment (SC1) · 4 Nov 2017

**Review of 'Assessing the impact of large volcanic eruptions of the Last Millenium on Australian rainfall regimes' by Blake et al.**

This paper is about the responses of the El Niño Southern Oscillation (ENSO), the Indian Ocean Dipole (IOD) and Australian precipitation to tropical volcanic eruptions. 9 ensembles from the NASA GISS ModelE2-R were analysed and run for the six largest tropical volcanic eruptions between 850 and 1850 CE. Anomalous conditions in ENSO, the IOD and Australian rainfall as a result of these volcanic eruptions were explored. Results show that large tropical eruptions during the last millennium indeed impact the large-scale IOD and ENSO systems and the Australian rainfall regimes. Larger mean atmospheric sulfate loading results in more persistent and more extreme positive IOD conditions and a stronger ENSO response. A positive response of Australian precipitation to volcanic forcing was found, although this response is stronger in NW Australia than in SE Australia.

The still relatively unclear relationships between tropical volcanic eruptions and the El Niño Southern Oscillation (ENSO), the Indian Ocean Dipole (IOD) and Australian precipitation were thus successfully explored with your research. Since you give a clear overview of these relationships and your approach can also be applied for exploring the impact of time-evolving forcings, such as volcanism, in other regions, this research strongly contributes to this field of research. Moreover, the paper fits the scope of the journal 'Climate of the Past', since the impact of historic volcanic eruptions on climatic variables is evaluated.

You start the paper with an elaborate introduction where a lot of references to previously published literature on this topic are made. You compare the results of several papers, which provides the reader already with some idea of the relationships that can be expected to be found in this paper and an overview of the state of the art of this field of research. A clear objective of the study is stated at the end of the introduction, which provides a concrete overview of the content of the paper. The results of the research are well-structured and have a logical order, since the results of all relationships between the variables are discussed one by one. Moreover, the figures of the results are clear and provide a good overview of all final results. The text in the results section and the result figures match and complement each other. The statements that are made in the discussion are well-funded on the results or on information from previous papers. I like the fact that differences between the results of this paper and of previous research are compared and possible explanations are given. Besides, the discussion and conclusion fit well to the relationships that were going to be explored as stated in the introduction, so the circle of the paper is closed. The paper contains a nice discussion on the limitations of the approach. Based on other literature some improvements are stated that should be made in future modelling in order to improve the accuracy of volcanic eruption model simulations. The improvements stated here form actually a small summary of previously published literature that can be consulted in order to figure out the exact adaptations that will improve the modelling of volcanic eruptions and corresponding processes.

In conclusion, I think your paper is well-written and a valuable contribution to this field of research. However, there are three major weaknesses that I think need to be solved before your paper can be published. These are explained below in this review and I also included some minor points that need to be improved in order to clarify some points in your paper.

**Major arguments**

1.  Your methodology contains in line 102 the statement that the Coupled Model Intercomparison Project Phase (CMIP5) is used in the NASA GISS ModelE2-R. However, Taylor et al. (2012) explain that the CMIP5 strategy can be used for long-term (century time scale) and near-term integrations (10-30 year). You explored anomalous conditions in the ENSO, the IOD and Australian rainfall for 7 years in total and only five years after a volcanic eruption, since this minimizes the effect of trends or low-frequency climate variability, which is a good argumentation. However, I wonder if the use of the CMIP5 gives reliable short-term model results for this short time period.
    Since you make you use of the NASA GISS ModelE2-R General Circulation Model and you refer to Schmidt et al. (2014) in line 100, I assume that your model contains all components that are taken into account by Schmidt et al. (2014) and that it includes an interactive representation of the atmosphere, ocean, land and sea ice. I expect that for most atmospheric processes the shorter timescale of your research will not be a problem, since most of these atmospheric processes are fast. The influence of aerosol injection into the atmosphere after a volcanic eruption will quickly have a noticeable influence in the model on for example atmospheric temperature, incoming shortwave solar radiation and cloud formation. However, ocean and land, domains that are also taken into account in the model, will have slower responses to volcanic eruptions. For example sea surface temperature, ocean currents and permafrost presence will take longer to adapt to the aerosol injection and corresponding atmospheric changes. For these variables the five year time scale that is investigated in your research might possibly be too short in order to explore the trend that occurs after a volcanic eruption.
    My recommendation is to validate the model results of these 7 year runs with available data and add these results to your paper. Is it a possibility to gather data of the ENSO, IOD and Australian rainfall anomalies for the times following the six eruptions used in you research and compare these with your model results? If this data is not available, because your eruptions occurred a long time ago, it might also be possible to use more recent data of the ENSO, IOD and Australian rainfall anomalies in years with more recent volcanic eruptions and compare these with new model results of these more recent volcanic eruptions. These volcanic eruptions are maybe smaller in size and have a smaller sulfate aerosol injection, but at least a validation of the model can be made in this way in order to check the use of CMIP5 for the relative short time period.

2.  I do not think that the methodology, mainly section 2.1 Simulations, contains enough information to understand your exact process in order to set up and make use of the model. Information that is missing is which data you used and what its source is, why you chose to make use of the specific NASA GISS ModelE2-R and the CMIP5, which variation of this model and which configuration were used and which values were for example used for the effective radius of the sulfate droplets. Schmidt et al. (2014) discuss different model configurations of the NASA GISS ModelE2-R, different ocean models and global annual mean features over the period 1980-2004 for the different models, which gives me the idea that you also made these kind of choices before you started modelling. Miller et al. (2014) show that there are three versions of the atmospheric model (NINT, TCADI and TCAD) which treat the atmospheric

constituents and the aerosol indirect effect differently. I assume that you also used one of these models, but it is not described which one you chose and why.

Moreover, you do not explain why you chose to analyse nine ensembles from the NASA GISS ModelE2-R, as stated in line 100, and not more or less. It is also not explained why five ensembles were forced with volcanic forcing, while four were not. Of the five run with volcanic forcing, four were forced with Crowley and Unterman (2013)'s aerosol optical depth data and one with double the Ice-core Volcanic Index 2 by Gao et al. (2008), but why are these not equally divided? Is it not more logical to force for example three ensembles with Crowley and Unterman (2013) and three with Gao et al. (2008)?

I would recommend to expand the methodology section of your paper with a more elaborate description of the exact methods. It will improve the paper if an overview of the steps that were taken is added, including the data that is used, model configurations and parameter values. Also an argumentation for the choices that were made will result in a more complete understanding of the methodology. A more extended discussion can also be added to the paragraph starting at line 264 then, discussing whether the chosen methodology turned out to be appropriate or if other choices should have been made.

3. A lot of relationships are stated in the introduction between tropical eruptions and the ENSO, IOD and Australian rainfall. For example, volcanism leads to negative global precipitation anomalies, large tropical eruptions can increase the likelihood and amplitude of an El Niño event in following years and a negative IOD occurs immediately after an eruption and a positive IOD one year later. For the relationship between volcanic eruptions and the ENSO, two possible mechanisms are mentioned in lines 49-53, although not in much detail, and the other relationships are not explained at all. However, you already refer to quite some papers that contain a more elaborate description behind the relationships.

If the mechanisms behind the processes would have been incorporated in the introduction, these mechanisms could also have been used in the discussion and conclusions section to explain the results that were found in your study. It could be checked whether the results correspond to these processes or if other processes are needed to explain the results. An example is that it would be useful if the processes that are taking place can be used to explain the difference between the timing of the peaks in figure 3 and 5, since this is currently not discussed in the paper.

There are two specific results mentioned that I think will definitely become more understandable if a discussion in which the processes are taken into account is added. Line 239 in the discussion and conclusions section tells us that tropical volcanism leads to positive precipitation anomalies over SE and NW Australia. In line 34 in the introduction it was stated, however, that sulfate aerosols result in negative global precipitation anomalies. I am puzzled by this contradiction, could it be caused by different processes that are occurring at different scales?

Moreover, in line 232-233 of the discussion it is stated that an El Niño-like pattern in the eastern Pacific is most visible in year 4, but also in year 0, 1 and 3. However, an explanation for this occurrence is not given, while I am wondering what occurred during year 2 that no El Niño was observed.

I would recommend to include a broader overview of all mechanisms behind the relationships in the introduction and take these mechanisms into account in the discussion of your results. The references in your paper about the mechanisms can be used for this adaptation, for example Clement et al. (1996), Mann et al. (2005), Pausata et al. (2015), Cheung & Abram (2016) and Meyers et al. (2007). Adding these explanations to your paper would really help the reader with understanding the physical processes and consequently the

relationships that are discussed. Besides, if these physical processes are more discussed in the introduction, they can also be used to explain the results of your research, for example the two specific results whose causes were unclear to me, as I mentioned above.

**Minor arguments**

1. A result of your research that is not mentioned in the abstract is that volcanic aerosol cooling dominates the precipitation response, while this is, in my opinion, an important result that should also be stated in the abstract. I would recommend to add this result to your abstract after the other results that are mentioned.

2. In the introduction the research question(s) is/are not clearly stated, although this would help with providing the reader with a better overview of the contents of the paper. I am also missing a broader aim of the paper and societal relevance, since it does not become completely clear what you actually want to achieve with your research and how you think your research will contribute to society. I would recommend to add the research question(s) and societal relevance to the introduction of the paper and the societal relevance might also be mentioned in the conclusion, stating what the results and conclusions can be used for.

3. In line 127-128 you explain why you chose to examine the IOD over the July-November period and you refer to Weller et al. (2014). However, their paper and also their results are only about positive IOD's. Negative IOD's are mentioned only twice in the whole paper, so I am not sure you can refer to this paper when your research examines both positive and negative IOD's. If you think you were right to still use the statement of Weller et al. (2014), I would like to see your explanation about this and otherwise you might consider taking into account a different period.

4. You state in line 134-136 the reason why you chose to analyse precipitation anomalies in southeast and northwest Australia and you refer to Ashok et al. (2004). However, most results of Ashok et al. (2004) are only about India, Pakistan and the monsoon trough and I do not find any mentioning of Australia. It would be good to check this reference and, if it turns out to be still the right one, to mention which results of their research you used. If the reference is not correct, please change the reference into the one that you based your statement on.

5. In line 133 in the methodology it is stated that the rainfall anomalies were examined over the July-November period, but it is not explained why you chose this period. It could be that the precipitation results are completely different during the other part of the year, for example that precipitation anomalies are negative instead of positive. I think it would be interesting to also take this into account in your research in order to have a more complete yearly overview, so maybe you could also perform these model simulations. Otherwise you could explain in your methodology why this was not necessary or possible.

6. You chose to do your research for the six largest tropical eruptions between 850-1850 CE as stated in line 114 in the methodology, while previous research, for example Cheung & Abram (2015) and Maher et al. (2015), took also smaller eruptions into account. Would it not be useful to also include these smaller eruptions in your research, since a better comparison between your study and these other studies can be made then? If you have a specific reason why you only chose the six largest eruptions, I would recommend to explain this reason in the methodology section after you mention which eruptions you analysed.

**Minor issues**

- Page 3, line 126: Please add a reference for the use of the NINO3.4 index and its calculation.
- Page 6, line 236: 'Increases' should be 'increase'.
- Page 15, caption figure 10: 'Mean SE' is missing in the caption of this figure if I compare it with the caption of figure 8, so please add this.

The rest of the paper is very well-structured and does not contain any mistakes.

---

## Author Comment (AC1) · 4 Dec 2017

Dear Reviewer,

Thank you for taking the time to read and comment on our paper. Your review was very helpful and we plan to make several important revisions in light of your recommendations.

There were three main issues in the paper that you identified: 1) Lack of information on experimental design – In our revision, we plan to expand the simulations section of the data and methods to include the following information: - The atmospheric model

is coupled with the dynamical Russel Ocean Model (Schmidt et al., 2014) - The atmospheric model was run with the NonINTeractive (NINT) atmospheric composition treatment (Schmidt et al., 2014) - The AOD was specified as per Crowley and Unterman (2012) or Gao et al. (2008)'s aerosol optical depth data with Reff specified as per Sato et al. (1993) – specifically this study used a 4 layer (15- 20km, 20-25km, 25-30km and 30-35km) vertical and 24 layer (8 degrees) latitude, longitudinally independent AOD with Reff specified as per Sato et al. (1993). 2) Lack of model evaluation – The GISS E2-R model has undergone evaluation in previous studies in reference to global precipitation and surface temperature (Dee et al., 2010; Flato et al., 2013; Schmidt et al., 2014), Australian precipitation and surface temperature (CSIRO and Bureau of Meteorology, 2015) and the ENSO and IOD (Flato et al., 2013; Schmidt et al., 2014). We plan to n include an additional literature review of these results in the simulations section as a justification of GISS's ability to simulate key climate aspects examined in this paper. 3) Lack of physical mechanisms – We will conduct further examination of the physical mechanisms, to contribute to our study's agreement with previous work in the results and discussion.

You specific remarks were also appreciated and most of these changes will be made. One exception to this, is your comment on table 1:

'It says strongest eruptions in the last millennium but I am pretty sure that Tambora was stronger than Huaynaputina. In fact, even the unknown 1809 eruption is bigger. Please adjust your table description.' - We plan to keep our table description as it is. Tambora is included in the table, and listed as the 3rd largest eruption, while Huaynaputina is listed as the 6th. The 1809 eruption was not larger than Huaynaputina, with a total global stratospheric sulfate loading (Tg) of 53.74, compared to Huaynaputina's 56.59. We presume that instead you are referring to the 1783 eruption, with a Tg of 92.96. We specify in the table title that we are looking at the largest 'tropical' eruptions, which are defined in the paper as eruptions that significantly impacted both hemispheres, as recorded by Gao et al., (2012), of which the 1783 eruption was not, with the entire

92.96 Tg limited to the northern hemisphere.

Once again, we thank you for your comments, and hope that you agree that our response has addressed your concerns.

Regards, Stephanie Blake, on behalf of the authors

---

## Author Comment (AC2) · 4 Dec 2017

Dear Reviewer,

Thank you for taking the time to read and comment on our paper. Your review was very helpful and we plan to make several important revisions in response to your recommendations.

There are five major concerns you outlined: 1) The set up and content of the introduction – In light of your comments, we agree that the Introduction could be more informative and structured. We plan to change the introduction to begin with an explanation of Australian rainfall, rather than the ENSO or IOD, and clarify the uncertainties in the relationship between volcanic aerosols and Australian precipitation. 2) Lack of model evaluation - The GISS E2-R model has undergone evaluation in previous studies in reference to global precipitation and surface temperature (Dee et al., 2010; Flato et al., 2013; Schmidt et al., 2014), Australian precipitation and surface temperature (CSIRO and Bureau of Meteorology, 2015) and the ENSO and IOD (Flato et al., 2013; Schmidt et al., 2014). We plan to include an additional literature review of these results in the simulations section as a justification of GISS's ability to simulate key climate aspects examined in this paper. 3) Lack of spatial figures dedicated to Australia – While we understand your concerns here, we believe the results from this paper will be of broader interest, so we have chosen to show global maps. We believe the maps are sufficiently detailed to show the Australian response, as well as the broader one, and thus, chose not to include maps dedicated solely to Australia. 4) Comparing the response of the largest and smallest eruptions – While a comparison of all 6 eruptions individually to the IOD, ENSO and Australia rainfall is feasible, we believe that providing the overall multi-ensemble multi-volcanic mean as well as the multi-ensemble mean for the largest and smallest eruptions individually was sufficient to support the arguments made in this paper. We are not arguing that larger eruptions will exponentially cause stronger responses in Australian rainfall, the IOD and ENSO, however we do argue that significantly larger eruptions e.g. Samalas (257.91 Tg) when compared to Huaynaptina (56.59 Tg), are more likely to cause a stronger and more persistent response, which the graphs provided, do show. We will modify the discussion section to clarify this aspect of the paper. 5) No examination of physical mechanisms - We shall conduct a further examination of the physical mechanisms, to contribute to our study's agreement with previous work in the results and discussion.

Your more specific remarks on the clarification of certain lines or statements were also noted and appreciated, and those changes will be made.

Once again, we thank you for your comments. We hope that you agree that our response has addressed your concerns.

Regards, Stephanie Blake, on behalf of the authors

---

## Author Comment (AC3) · 4 Dec 2017

Dear Marte Stoorvogel,

Thank you for taking the time to read and comment on our paper. Your review was very helpful and we plan to make several important revisions in light of your recommendations.

You listed several concerns that we believe warrant an in-depth response: 1) Suitability of timescales used for analysis of land and ocean domains – While we understand your concerns that the effect of volcanic aerosols have a delayed response in the ocean and

land domains when compared to the atmosphere, we believe these concerns are irrelevant to the scope of this paper. There is extensive literature that supports the theory that precipitation, surface temperature, the ENSO and the IOD all only show a statistical response to volcanic aerosols within the first five years post eruption (Adams et al., 2003; Cheung & Abram., 2016; Emile-Geay et al., 2008; Gillett et al., 2004; Illes et al., 2013; Joseph and Zeng et al., 2011; Maher et al., 2015; Mann et al., 2005; McGregor et al., 2010; Pausata et al., 2014; Predybaylo et al., 2017; Schneider et al., 2009; Soden et al., 2002; Wahl et al., 2014) . While ocean currents and the deep ocean do take longer to respond, these processes are not examined in this paper. We therefore consider it unnecessary to alter the timescales used in our analysis. 2) Lack of information on experimental design – You raised a valid point here, and we plan to expand the simulations section of the data and methods on revision to include the following information: - The atmospheric model is coupled with the dynamical Russel Ocean Model (Schmidt et al., 2014) - The atmospheric model was run with the NonINTeractive (NINT) atmospheric composition treatment (Schmidt et al., 2014) - The AOD was specified as per Crowley and Unterman (2012) or Gao et al. (2008)'s aerosol optical depth data with Reff specified as per Sato et al. (1993) – specifically this study used a 4 layer (15-20km, 20-25km, 25-30km and 30-35km) vertical and 24 layer (8 degrees) latitude, longitudinally independent AOD with Reff specified as per Sato et al. (1993). 3) Choice of forcing – Our choice of the combination of 5 volcanically forced ensembles, and 4 non-volcanically forced ensembles was driven by a desire to see a comparison of the effect of volcanic aerosols with non-volcanically influenced scenarios to strengthen any arguments for the effect of aerosols. 4) No explanation of physical mechanisms - We shall conduct further examination of the physical mechanisms, to contribute to our study's agreement with previous work in the results and discussion.

Your other, smaller comments on clarifications and references have been noted and will be changed accordingly.

We thank you for your comments and we hope you agree that our response has addressed your concerns.

Regards, Stephanie Blake, on behalf of all authors
* * *

---

## Author Response (AR1)

**1) Comments from referees/public:**

**Anonymous Referee #1**

5 Received and published: 3 October 2017

GENERAL REMARKS:

- The manuscript "Assessing the impact of large volcanic eruptions of the Last Millennium on Australian rainfall 10 regimes" in consideration for publication in Climate of the Past investigates the impacts of tropical explosive volcanic eruptions on Australian rainfall regimes, ENSO, and the Indian Ocean Dipole. The authors use a set of specific global climate model realisations to address questions about the importance of volcanic forcing power and timing of the above mentioned impacts.
- 15 Generally, the topic of the manuscript fits Climate of the Past and is a valuable contribution to the community. The study sheds light on volcanic climate impacts in the Southern Hemisphere, for which only few studies exist so far.

However, I feel like the manuscript lacks content in some important parts. In my opinion, the three main weaknesses are:

20

25

30

- 1) Experimental design: Since the study largely focuses on the IOD and ENSO, the authors should describe in detail how SSTs and evaporation over oceans is handled in the model. Is it AMIP run? Is it coupled? Slab ocean? And if the SSTs are prescribed, can it be used at all? These things are completely disregarded in the manuscript at hand.
- 2) Model evaluation: There is NO effort made to convince me that this model is doing a good job in terms of precipitation. I would like to see a comparison between model and proxy data, or model and (PDSI) reconstructions.
  - 3) Physical mechanisms behind the shown effect on ENSO and IOD: Although a lot of studies are cited as to show that there is agreement with previous work, I didn't find a proper physical explanation for the effects we see in the plots besides the paragraph in the introduction. I would like to see a discussion about the mechanisms in this specific model.

See below for specific remarks.

**35 SPECIFIC REMARKS:**

Line 34: I would add here newer studies about precipitation like Iles et al 2013 or Wegmann et al 2014

Line 93: Not sure what exactly is unclear? Do you mean dynamic vs radiative changes? Please explain further what gap 40 your study is filling.

Line 103: Here I really would like to see a thorough description of the ocean setting for this model (see above). What kind of SSTs are used? Is there a dynamic ocean? How is the volcanic signal supposed to show in the SSTs? Is it valid to use the model to investigate ocean surface changes? CMIP5 also had AMIP run included, so I can't really infer if your model was coupled or not.

45

Line 103: I assume CO2 forcing is done in CMIP5 fashion?

Line 148: As I said before, I would like to see an evaluation for starters. How good is the model in terms of 50 precipitation? The comparison doesn't have to be for Australia, but I want to be convinced that the model gets the broader precipitation response. Otherwise, the rest of the results is less meaningful.

Figure 2: Is this the annual mean? If so please indicate the fact. Maybe it would be nice to see DJF and JJA anomalies? Is the model able to do the NH winter warming? And if not, is there an argument to make that the model doesn't get the dynamics right (as is the case with many CMIP5 models)?

Line 176: Again, I wonder about NH winter warming. Should counteract the summer cooling over the NH.

Line 194: Indeed it suggests that. But how does it work? Where is the discussion of the physical mechanisms? How is 60 the signal transported in the model? Maybe show evaporation, heat content and other metrics to show the mechanisms in the model.

Line 241: Okay, but how does this effect override the impact of IOD?

Table 1: It says strongest eruptions in the last millennium but I am pretty sure that Tambora was stronger than Huaynaptina as Figure 1 shows. In fact even the unknown 1809 eruption is bigger. Please adjust your table description.

Figure 3: In fact 2xCG is not an ensemble and should not be marked as such.

70 Figure 4: The 90th percentile of what? The ensemble members? If so, please adjust. (also for the rest of the Samalas & Huaynaputina plots.

Figure 10: Here it says Australian precipitation whereas Figure 9 says NW Australian precipitation and Figure 11 says SE Australian precipitation. I assume Figure 10 also is SE precipitation?

**Anonymous Referee #2**

Received and published: 12 October 2017

80 This work analysed the precipitation response to large volcanic perturbation in Australian, using GISS Model E millennium simulations. The topic is an important contribution to the systematic assessment of global hydroclimate response to the volcanic radiative influence. The paper is well written; the results are clearly presented and discussed. I would recommend a major revision of the paper by addressing the following issues, before it could be accepted for publication:

85

90

95

75

55

- In the "Introduction" section 1.1 It is not clear from the description which relationships between volcanic eruptions and Australian rainfall are unclear and are explored in this paper. Please be more specific. A significant part of the introduction is devoted to the literature review of the volcanism and ENSO relationship, while the focus of this study is on volcanic eruptions and Australian rainfall. Please modify either the title, or the structure of introduction. For example, the authors may consider introduce the role of ENSO in changing Australian rainfall before discuss the volcanism and ENSO. Relationship
- 2. In the "Data and Methods" section 2.1. Line 99, please explain briefly why the last millennium period is especially chosen. 2.2 Please provide a short description of model performance, especially those closely related to the ENSO, IOD and volcanic climate responses. Please also provide a short explanation of why the five volcanically forced scenarios were chosen. 2.3 Line 134, please also discuss the ENSO impact in the chosen north-west and south-east regions.
- In the "Results" section 3.1 Please explain the advantages of comparing the response between the largest 1257
   Samalas and the smallest 1600 Huaynaptina eruption, rather than a series comparison among the 6 eruptions of various size. 3.2 Line 164, there is no ensemble runs for the 2xGC case, please make the distinction. 3.3 One of Figure 8 or Figure 9 should be "SE (instead of NW) Australia" response. Please also correct the reference to the Figure 10 in Line 179. The use of "multi-model mean" in several figures is misleading, please consider

135

change to model ensemble. 3.4 The focus of this work is Australia, however, there is no spatial figures dedicated to the particular area of study. 3.5 Please verify the use of 0.6 standard deviation as the significant level.

4. In the "Discussion and Conclusions" section 4.1 line 218-221, the difference between Samalas and Huaynaptina response does not seem significant and the use of it as support for the persistence of a high pIOD 110 sounds weak to me. Please consider the use of all six (or even more eruptions) of different sizes to analyse the role of eruption magnitude. 4.2 Line 228-235, I do not see the El Nino response persist from year 0 to year 6 in both Fig. 5 and Fig. 2. Please explain why the El Nino-like pattern in the eastern Pacific is most visible in year 4, but not earlier. 4.3 Line 241-243, please demonstrate in more detail how did the direct precipitation effect of volcanic aerosols override the impact of the IOD on Australian precipitation. Which parameter represents the 115 direct precipitation effect of volcanic aerosols? How was the override effect appear in the results? By the time difference? 4.4 The most important advantage of using modelling results is the capacity of exam the physical mechanisms behind the shown effects (such as the impact on ENSO and IOD, and their influence on Australian rainfall). Please provide some discussion of mechanism using the original results from this paper, rather than referring to previous studies. 120

**Public Review #1:**

This paper is about the responses of the El Niño Southern Oscillation (ENSO), the Indian Ocean Dipole (IOD) and Australian precipitation to tropical volcanic eruptions. 9 ensembles from the NASA GISS ModelE2-R were analysed 125 and run for the six largest tropical volcanic eruptions between 850 and 1850 CE. Anomalous conditions in ENSO, the IOD and Australian rainfall as a result of these volcanic eruptions were explored. Results show that large tropical eruptions during the last millennium indeed impact the large-scale IOD and ENSO systems and the Australian rainfall regimes. Larger mean atmospheric sulfate loading results in more persistent and more extreme positive IOD conditions and a stronger ENSO response. A positive response of Australian precipitation to volcanic forcing was found, although this response is stronger in NW Australia than in SE Australia.

130

eruptions on climatic variables is evaluated.

The still relatively unclear relationships between tropical volcanic eruptions and the El Niño Southern Oscillation (ENSO), the Indian Ocean Dipole (IOD) and Australian precipitation were thus successfully explored with your research. Since you give a clear overview of these relationships and your approach can also be applied for exploring the impact of time-evolving forcings, such as volcanism, in other regions, this research strongly contributes to this field of research. Moreover, the paper fits the scope of the journal 'Climate of the Past', since the impact of historic volcanic

You start the paper with an elaborate introduction where a lot of references to previously published literature on this topic are made. You compare the results of several papers, which provides the reader already with some idea of the

- relationships that can be expected to be found in this paper and an overview of the state of the art of this field of 140 research. A clear objective of the study is stated at the end of the introduction, which provides a concrete overview of the content of the paper. The results of the research are well-structured and have a logical order, since the results of all relationships between the variables are discussed one by one. Moreover, the figures of the results are clear and provide a good overview of all final results. The text in the results section and the result figures match and complement each other. The statements that are made in the discussion are well-funded on the results or on information from previous
- 145 papers. I like the fact that differences between the results of this paper and of previous research are compared and possible explanations are given. Besides, the discussion and conclusion fit well to the relationships that were going to be explored as stated in the introduction, so the circle of the paper is closed. The paper contains a nice discussion on the

limitations of the approach. Based on other literature some improvements are stated that should be made in future modelling in order to improve the accuracy of volcanic eruption model simulations. The improvements stated here form actually a small summary of previously published literature that can be consulted in order to figure out the exact adaptations that will improve the modelling of volcanic eruptions and corresponding processes.

In conclusion, I think your paper is well-written and a valuable contribution to this field of research. However, there are three major weaknesses that I think need to be solved before your paper can be published. These are explained below in this review and I also included some minor points that need to be improved in order to clarify some points in your paper.

**MAJOR ARGUEMENTS**

150

155

- 1. Your methodology contains in line 102 the statement that the Coupled Model Intercomparison Project Phase (CMIP5) is used in the NASA GISS ModelE2-R. However, Taylor et al. (2012) explain that the CMIP5 strategy can be used for long-term (century time scale) and near-term integrations (10-30 year). You explored 160 anomalous conditions in the ENSO, the IOD and Australian rainfall for 7 years in total and only five years after a volcanic eruption, since this minimizes the effect of trends or low-frequency climate variability, which is a good argumentation. However, I wonder if the use of the CMIP5 gives reliable short-term model results for this short time period. Since you make you use of the NASA GISS ModelE2-R General Circulation Model and you refer to Schmidt et al. (2014) in line 100, I assume that your model contains all components that are 165 taken into account by Schmidt et al. (2014) and that it includes an interactive representation of the atmosphere, ocean, land and sea ice. I expect that for most atmospheric processes the shorter timescale of your research will not be a problem, since most of these atmospheric processes are fast. The influence of aerosol injection into the atmosphere after a volcanic eruption will quickly have a noticeable influence in the model on for example atmospheric temperature, incoming shortwave solar radiation and cloud formation. However, ocean and land, 170 domains that are also taken into account in the model, will have slower responses to volcanic eruptions. For example sea surface temperature, ocean currents and permafrost presence will take longer to adapt to the aerosol injection and corresponding atmospheric changes. For these variables the five year time scale that is investigated in your research might possibly be too short in order to explore the trend that occurs after a volcanic eruption. My recommendation is to validate the model results of these 7 year runs with available data 175 and add these results to your paper. Is it a possibility to gather data of the ENSO, IOD and Australian rainfall anomalies for the times following the six eruptions used in you research and compare these with your model results? If this data is not available, because your eruptions occurred a long time ago, it might also be possible to use more recent data of the ENSO, IOD and Australian rainfall anomalies in years with more recent volcanic eruptions and compare these with new model results of these more recent volcanic eruptions. These 180 volcanic eruptions are maybe smaller in size and have a smaller sulfate aerosol injection, but at least a validation of the model can be made in this way in order to check the use of CMIP5 for the relative short time period.
- I do not think that the methodology, mainly section 2.1 Simulations, contains enough information to understand your exact process in order to set up and make use of the model. Information that is missing is which data you used and what its source is, why you chose to make use of the specific NASA GISS ModelE2-R and the CMIP5, which variation of this model and which configuration were used and which values were for example used for

the effective radius of the sulfate droplets. Schmidt et al. (2014) discuss different model configurations of the NASA GISS ModelE2-R, different ocean models and global annual mean features over the period 1980-2004 for the different models, which gives me the idea that you also made these kind of choices before you started 190 modelling. Miller et al. (2014) show that there are three versions of the atmospheric model (NINT, TCADI and TCAD) which treat the atmospheric constituents and the aerosol indirect effect differently. I assume that you also used one of these models, but it is not described which one you chose and why. Moreover, you do not explain why you chose to analyse nine ensembles from the NASA GISS ModelE2-R, as stated in line 100, and not more or less. It is also not explained why five ensembles were forced with volcanic forcing, while four 195 were not. Of the five run with volcanic forcing, four were forced with Crowley and Unterman (2013)'s aerosol optical depth data and one with double the Ice-core Volcanic Index 2 by Gao et al. (2008), but why are these not equally divided? Is it not more logical to force for example three ensembles with Crowley and Unterman (2013) and three with Gao et al. (2008)? I would recommend to expand the methodology section of your paper with a more elaborate description of the exact methods. It will improve the paper if an overview of the steps 200 that were taken is added, including the data that is used, model configurations and parameter values. Also an argumentation for the choices that were made will result in a more complete understanding of the methodology. A more extended discussion can also be added to the paragraph starting at line 264 then, discussing whether the chosen methodology turned out to be appropriate or if other choices should have been made.

- A lot of relationships are stated in the introduction between tropical eruptions and the ENSO, IOD and Australian rainfall. For example, volcanism leads to negative global precipitation anomalies, large tropical eruptions can increase the likelihood and amplitude of an El Niño event in following years and a negative IOD occurs immediately after an eruption and a positive IOD one year later. For the relationship between volcanic eruptions and the ENSO, two possible mechanisms are mentioned in lines 49-53, although not in much detail, and the other relationships are not explained at all. However, you already refer to quite some papers that contain a more elaborate description behind the relationships.
- If the mechanisms behind the processes would have been incorporated in the introduction, these mechanisms could also have been used in the discussion and conclusions section to explain the results that were found in your study. It could be checked whether the results correspond to these processes or if other processes are needed to explain the results. An example is that it would be useful if the processes that are taking place can be used to explain the difference between the timing of the peaks in figure 3 and 5, since this is currently not discussed in the paper.
- There are two specific results mentioned that I think will definitely become more understandable if a discussion in which the processes are taken into account is added. Line 239 in the discussion and conclusions section tells us that tropical volcanism leads to positive precipitation anomalies over SE and NW Australia. In line 34 in the introduction it was stated, however, that sulfate aerosols result in negative global precipitation anomalies. I am puzzled by this contradiction, could it be caused by different processes that are occurring at different scales? Moreover, in line 232-233 of the discussion it is stated that an El Niño-like pattern in the eastern Pacific is most visible in year 4, but also in year 0, 1 and 3. However, an explanation for this occurrence is not given, while I am wondering what occurred during year 2 that no El Niño was observed. I would recommend to include a broader overview of all mechanisms behind the relationships in the introduction and take these mechanisms into account in the discussion of your results. The references in your

paper about the mechanisms can be used for this adaptation, for example Clement et al. (1996), Mann et al. (2005), Pausata et al. (2015), Cheung & Abram (2016) and Meyers et al. (2007). Adding these explanations to your paper would really help the reader with understanding the physical processes and consequently the relationships that are discussed. Besides, if these physical processes are more discussed in the introduction, they can also be used to explain the results of your research, for example the two specific results whose causes were unclear to me, as I mentioned above.

**MINOR ARGUEMENTS**

230

255

- 235 1. A result of your research that is not mentioned in the abstract is that volcanic aerosol cooling dominates the precipitation response, while this is, in my opinion, an important result that should also be stated in the abstract. I would recommend to add this result to your abstract after the other results that are mentioned.
- 2. In the introduction the research question(s) is/are not clearly stated, although this would help with providing the reader with a better overview of the contents of the paper. I am also missing a broader aim of the paper and societal relevance, since it does not become completely clear what you actually want to achieve with your research and how you think your research will contribute to society. I would recommend to add the research question(s) and societal relevance to the introduction of the paper and the societal relevance might also be mentioned in the conclusion, stating what the results and conclusions can be used for.
- 3. In line 127-128 you explain why you chose to examine the IOD over the July-November period and you refer to Weller et al. (2014). However, their paper and also their results are only about positive IOD's. Negative IOD's are mentioned only twice in the whole paper, so I am not sure you can refer to this paper when your research examines both positive and negative IOD's. If you think you were right to still use the statement of Weller et al. (2014), I would like to see your explanation about this and otherwise you might consider taking into account a different period.
- 4. You state in line 134-136 the reason why you chose to analyse precipitation anomalies in southeast and northwest Australia and you refer to Ashok et al. (2004). However, most results of Ashok et al. (2004) are only about India, Pakistan and the monsoon trough and I do not find any mentioning of Australia. It would be good to check this reference and, if it turns out to be still the right one, to mention which results of their research you used. If the reference is not correct, please change the reference into the one that you based your statement on.
- 5. In line 133 in the methodology it is stated that the rainfall anomalies were examined over the July-November period, but it is not explained why you chose this period. It could be that the precipitation results are completely different during the other part of the year, for example that precipitation anomalies are negative instead of positive. I think it would be interesting to also take this into account in your research in order to have a more complete yearly overview, so maybe you could also perform these model simulations. Otherwise you could explain in your methodology why this was not necessary or possible.
  - 6. You chose to do your research for the six largest tropical eruptions between 850-1850 CE as stated in line 114 in the methodology, while previous research, for example Cheung & Abram (2015) and Maher et al. (2015), took also smaller eruptions into account. Would it not be useful to also include these smaller eruptions in your

265 research, since a better comparison between your study and these other studies can be made then? If you have a specific reason why you only chose the six largest eruptions, I would recommend to explain this reason in the methodology section after you mention which eruptions you analysed.

**MINOR ISSUES**

Page 3, line 126: Please add a reference for the use of the NINO3.4 index and its calculation.

- 270 Page 6, line 236: 'Increases' should be 'increase'.
  - Page 15, caption figure 10: 'Mean SE' is missing in the caption of this figure if I compare it with the caption of figure 8, so please add this. The rest of the paper is very well-structured and does not contain any mistakes.

275

305

**310 **2)** Author's Response:**

Dear Reviewers,

315 Thank you for taking the time to read and comment on our paper. Your reviews were very helpful and we have now made several important revisions in light of these recommendations.

Anonymous Referee #1

- 320 1) Experimental design: Since the study largely focuses on the IOD and ENSO, the authors should describe in detail how SSTs and evaporation over oceans is handled in the model. Is it AMIP run? Is it coupled? Slab ocean? And if the SSTs are prescribed, can it be used at all? These things are completely disregarded in the manuscript at hand.
- We have addressed this concern at line 634-635. The atmospheric model is coupled with the dynamical Russel Ocean Model, and a reference (Schmidt et al., 2014) provided for further detail on the Russel Ocean Model configuration.

2) Model evaluation: There is NO effort made to convince me that this model is doing a good job in terms of precipitation. I would like to see a comparison between model and proxy data, or model and (PDSI) reconstructions.

- The GISS E2-R model has previously undergone comprehensive evaluation for rainfall metrics and drivers in previous studies (Flato et al., 2013; Schmidt et al., 2014; Moise et al., 2015; and Miller et al., 2015). We have now included a literature review of these studies covering surface temperature, precipitation, volcanic aerosols, the ENSO and the IOD (line 637-651) and conducted additional comparisons of modelled and observed NINO3 index intensity, seasonality and regression against SST and precipitation (Line 652-682 and Figure 1). We have not included a comparison of the model output with the PDSI as we do not believe that this provides greater insight into the model skills than investigating
- precipitation metrics directly.

3) Physical mechanisms behind the shown effect on ENSO and IOD: Although a lot of studies are cited as to show that there is agreement with previous work, I didn't find a proper physical explanation for the effects we see in the plots besides the paragraph in the introduction. I would like to see a discussion about the mechanisms in this specific model.

We have conducted further examination of the physical mechanisms driving the response of the IOD (line 863-870) and ENSO (line 876-881) in GISS to contribute to our studies agreement with previous work in the results and discussion, and included a description of the anomalous wind direction (Fig. 13) and speed (Fig. 14) following eruptions to

contribute to the IOD and ENSO response (line 807-812).

340

Line 34: I would add here newer studies about precipitation like Iles et al 2013 or Wegmann et al 2014

350 The Illes et al. (2013) reference was reviewed and added here (line 554).

Line 93: Not sure what exactly is unclear? Do you mean dynamic vs radiative changes? Please explain further what gap your study is filling.

355 We have modified the structure of the introduction to further outline the premise of this study: to analyse the impact of volcanic forcing on Australian rainfall due to its direct effect and the feedback effects of ENSO and the IOD. The lack of clarity on this subject is present due to lack of literature focused specifically on it (line 556-557 and line 623-624).

Figure 2: Is this the annual mean? If so please indicate the fact. Maybe it would be nice to see DJF and JJA anomalies?

360

380

This is the July-September mean (JASON). Clarification has been provided in figure titles (line 1156, 1191, 1229, 1231)

 Table 1: It says strongest eruptions in the last millennium but I am pretty sure that Tambora was stronger than
 365

 Huaynaptina as Figure 1 shows. In fact even the unknown 1809 eruption is bigger. Please adjust your table description.

We plan to keep our table description as it is. Tambora is included in the table, and listed as the 3rd largest eruption, while Huaynaputina is listed as the 6th. The 1809 eruption was not larger than Huaynaputina, with a total global stratospheric sulfate loading (Tg) of 53.74, compared to Huaynaputina's 56.59. We presume that instead referee #1 was referring to the 1783 eruption, with a Tg of 92.96. We specify in the table title that we are looking at the largest

- 370 referring to the 1783 eruption, with a Tg of 92.96. We specify in the table title that we are looking at the largest 'tropical' eruptions, which are defined in the paper as eruptions that significantly impacted both hemispheres, as recorded by Gao et al., (2012), of which the 1783 eruption was not, with the entire 92.96 Tg limited to the northern hemisphere.
- 375 Figure 3: In fact 2xCG is not an ensemble and should not be marked as such.

We agree, and this clarification has been added to all figures.

Figure 4: The 90th percentile of what? The ensemble members? If so, please adjust. (also for the rest of the Samalas and Huaynaputina plots.

It is the 90th and 10th percentile of the ensemble members. Clarification has been provided in figure titles (line 1173, 1188, 1206, 1220)

385 Figure 10: Here it says Australian precipitation whereas Figure 9 says NW Australian precipitation and Figure 11 says SE Australian precipitation. I assume Figure 10 also is SE precipitation?

This figure was meant to say SE, this alteration has been made (line 1212).

**390 Anonymous Referee #2**

In the "Introduction" section 1.1 It is not clear from the description which relationships between volcanic eruptions and Australian rainfall are unclear and are explored in this paper. Please be more specific. A significant part of the

introduction is devoted to the literature review of the volcanism and ENSO relationship, while the focus of this study is

- 395 on volcanic eruptions and Australian rainfall. Please modify either the title, or the structure of introduction. For example, the authors may consider introduce the role of ENSO in changing Australian rainfall before discuss the volcanism and ENSO Relationship
- We have improved the structure of the introduction, now beginning with a description of Australian rainfall 400 characteristics, before introducing the relationship between volcanic aerosols and ENSO and IOD systems. We have also clarified that the uncertainties in the relationship between volcanic aerosols and Australian rainfall lie in the lack of research devoted solely to the subject and due to the contrasting precipitation surplus impact of the direct radiative effect of aerosols, with the precipitation supressing impact of generated El Nino and positive IOD phases (line 554-558).
- 405

In the "Data and Methods" section 2.1. Line 99, please explain briefly why the last millennium period is especially chosen. 2.2 Please provide a short description of model performance, especially those closely related to the ENSO, IOD and volcanic climate responses. Please also provide a short explanation of why the five volcanically forced scenarios were chosen. 2.3 Line 134, please also discuss the ENSO impact in the chosen north-west and south-east regions.

410

Clarifications have been included to the data and methods section in light of these comments. A short literature review of previous model evaluation of GISS E2-R and further evaluations conducted on ENSO have been included (line 637-682 and Figure 1), and the reason for choosing the Last Millennium as a target period (due to a large number of recorded volcanic eruptions) has been outlined (line 688-689).

415

In the "Results" section 3.1 Please explain the advantages of comparing the response between the largest 1257 Samalas and the smallest 1600 Huaynaptina eruption, rather than a series comparison among the 6 eruptions of various size.

- 420 While a comparison of all 6 eruptions individually to the IOD, ENSO and Australia rainfall is possible, we believe that providing the overall multi-ensemble multi-volcanic mean as well as the multi-ensemble mean for the largest and smallest eruptions individually was sufficient to support the arguments made in this paper. We are not arguing that larger eruptions will exponentially cause stronger responses in the IOD and ENSO, however we do argue that significantly larger eruptions e.g. Samalas (257.91 Tg) when compared to Huaynaptina (56.59 Tg), are more likely to 425
- cause a stronger and more persistent response, which the graphs provided, do show.

3.4 The focus of this work is Australia, however, there is no spatial figures dedicated to the particular area of study. 3.5 Please verify the use of 0.6 standard deviation as the significant level.

**430**

While we understand this concern, we believe the results from our paper will be of broader interest, so we have chosen to show global maps. These maps are sufficiently detailed to show the Australian response, as well as the broader one, and thus, chose not to include maps dedicated solely to Australia.

435 4.2 Line 228-235, I do not see the El Nino response persist from year 0 to year 6 in both Fig. 5 and Fig. 2. Please

explain why the El Nino-like pattern in the eastern Pacific is most visible in year 4, but not earlier.

Evaluation of the physical mechanisms driving the more visible El Nino-like anomaly seem in Fig 3 in year 4 have been included in the discussion (line 876).

**440**

4.4 The most important advantage of using modelling results is the capacity of exam the physical mechanisms behind the shown effects (such as the impact on ENSO and IOD, and their influence on Australian rainfall). Please provide some discussion of mechanism using the original results from this paper, rather than referring to previous studies.

445 We have conducted further examination of the physical mechanisms driving the response of the IOD (line 863-870) and ENSO (line 876-881) in GISS to contribute to our studies agreement with previous work in the results and discussion, and included a description of the anomalous wind direction (Fig. 13) and speed (Fig. 14) following eruptions to contribute to the IOD and ENSO response (line 808-812).

**450 **Public Reviewer #1:**

Your methodology contains in line 102 the statement that the Coupled Model Intercomparison Project Phase (CMIP5) is used in the NASA GISS ModelE2-R. However, Taylor et al. (2012) explain that the CMIP5 strategy can be used for long-term (century time scale) and near-term integrations (10-30 year). You explored anomalous conditions in the

- 455 ENSO, the IOD and Australian rainfall for 7 years in total and only five years after a volcanic eruption, since this minimizes the effect of trends or low-frequency climate variability, which is a good argumentation. However, I wonder if the use of the CMIP5 gives reliable short-term model results for this short time period. Since you make you use of the NASA GISS ModelE2-R General Circulation Model and you refer to Schmidt et al. (2014) in line 100, I assume that your model contains all components that are taken into account by Schmidt et al. (2014) and that it includes an
- 460 interactive representation of the atmosphere, ocean, land and sea ice. I expect that for most atmospheric processes the shorter timescale of your research will not be a problem, since most of these atmospheric processes are fast. The influence of aerosol injection into the atmosphere after a volcanic eruption will quickly have a noticeable influence in the model on for example atmospheric temperature, incoming shortwave solar radiation and cloud formation. However, ocean and land, domains that are also taken into account in the model, will have slower responses to volcanic
- 465 eruptions. For example sea surface temperature, ocean currents and permafrost presence will take longer to adapt to the aerosol injection and corresponding atmospheric changes. For these variables the five year time scale that is investigated in your research might possibly be too short in order to explore the trend that occurs after a volcanic eruption.
- 470 While we understand these concerns that the effect of volcanic aerosols have a delayed response in the ocean and land domains when compared to the atmosphere, we believe these concerns are outside the scope of this paper. There is extensive literature that supports the view that precipitation, surface temperature, the ENSO and the IOD all only show a statistically significant response to volcanic aerosols within the first five years post eruption (Adams et al., 2003; Cheung & Abram., 2016; Emile-Geay et al., 2008; Gillett et al., 2004; Illes et al., 2013; Joseph and Zeng et al., 2011;
- 475 Maher et al., 2015; Mann et al., 2005; McGregor et al., 2010; Pausata et al., 2014; Predybaylo et al., 2017; Schneider et al., 2009; Soden et al., 2002; Wahl et al., 2014). While ocean currents and the deep ocean do take longer to respond,

these processes are not examined in this paper. We therefore consider it unnecessary to alter the timescales used in our analysis, though would consider this comment for future research.

- 480 I do not think that the methodology, mainly section 2.1 Simulations, contains enough information to understand your exact process in order to set up and make use of the model. Information that is missing is which data you used and what its source is, why you chose to make use of the specific NASA GISS ModelE2-R and the CMIP5, which variation of this model and which configuration were used and which values were for example used for the effective radius of the sulfate droplets.
- 485

We have addressed this concern at line 634-635. The atmospheric model was run with the Non-Interactive (NINT) atmospheric composition and is coupled with the dynamical Russel Ocean Model, and a reference (Schmidt et al., 2014) provided for further detail on both model configurations. A small literature review of previous evaluation of GISS E2-R and a personally conducted evaluation of ENSO is also provided to justify the use of the model in these studies (line 637-682) and the configuration of volcanic aerosols in the model described (line 689-691).

490

It is also not explained why five ensembles were forced with volcanic forcing, while four were not. Of the five run with volcanic forcing, four were forced with Crowley and Unterman (2013)'s aerosol optical depth data and one with double the Ice-core Volcanic Index 2 by Gao et al. (2008), but why are these not equally divided? Is it not more logical to force for example three ensembles with Crowley and Unterman (2013) and three with Gao et al. (2008)? I would recommend to expand the methodology section of your paper with a more elaborate description of the exact methods.

Our choice of the combination of 5 volcanically forced ensembles, and 4 non-volcanically forced ensembles was driven by a desire to see a comparison of the effect of volcanic aerosols with non-volcanically influenced scenarios to strengthen any arguments for the effect of aerosols (line 685). The use of more Crowley and Unterman AOD is due to

500 strengthen any arguments for the effect of aerosols (line 685). The use of more Crowley and U the fact that Gao et al. (2008)'s data was multiplied by 2 (line 687).

I would recommend to include a broader overview of all mechanisms behind the relationships in the introduction and take these mechanisms into account in the discussion of your results. The references in your paper about the

- 505 mechanisms can be used for this adaptation, for example Clement et al. (1996), Mann et al. (2005), Pausata et al. (2015), Cheung & Abram (2016) and Meyers et al. (2007). Adding these explanations to your paper would really help the reader with understanding the physical processes and consequently the relationships that are discussed. Besides, if these physical processes are more discussed in the introduction, they can also be used to explain the results of your research.
- 510

Examination of the mechanisms driving the response in GISS was undertaken, and provided explanation for responses seen in the model results to compliment the mechanisms described by the referenced paper (line 863-870, 876-881, 807-812).

12

**3) Changes to manuscript**

520

**Assessing the impact of large volcanic eruptions of the Last Millennium on Australian rainfall regimes**

| 1   | Stephanie A. P. Blake a,b , Sophie C. Lewis b,c , Allegra N. LeGrande d and Ron L. Miller d |           | Deleted: and                                                                                                                                                            |
|-----|-----------------------------------------------------------------------------------------------------------------------------------------|-----------|-------------------------------------------------------------------------------------------------------------------------------------------------------------------------|
| 525 |                                                                                                                                         | $\leq$    | Formatted: Superscript                                                                                                                                                  |
|     | a Climate Change Research Centre, University of New South Wales, Sydney, UNSW, Australia                                     |           |                                                                                                                                                                         |
|     | c Fenner School of Environment and Society, The Australian National University, Canberra, ACT, Australia                     |           |                                                                                                                                                                         |
|     | d NASA Goddard Institute for Space Studies and Center for Climate Systems Research, Columbia University                      |           |                                                                                                                                                                         |
| 530 | Corresponding author: Stephanie Blake (stephanieblake79@gmail.com)                                                                      |           |                                                                                                                                                                         |
|     |                                                                                                                                         |           |                                                                                                                                                                         |
|     | Abstract. Explosive volcanism is an important natural climate forcing, impacting global surface temperatures and                        |           |                                                                                                                                                                         |
|     | regional precipitation. Although previous studies have investigated aspects of the impact of tropical volcanism on                      |           |                                                                                                                                                                         |
|     | various ocean-atmosphere systems and regional climate regimes, volcanic eruptions remain a poorly understood climate                    |           |                                                                                                                                                                         |
| 535 | forcing and climatic responses are not well constrained. In this study, volcanic eruptions are explored in particular                   |           |                                                                                                                                                                         |
|     | reference to Australian precipitation, and both the Indian Ocean Dipole (IOD) and El Nino-Southern Oscillation                          |           |                                                                                                                                                                         |
|     | (ENSO). Using nine realisations of the Last Millennium (LM) with different time-evolving forcing combinations, from                     |           |                                                                                                                                                                         |
|     | the NASA GISS ModelE2-R, the impact of the 6 largest tropical volcanic eruptions of this period are investigated.                       |           |                                                                                                                                                                         |
|     | Overall, we find that volcanic aerosol forcing increased the likelihood of El Nino and positive IOD conditions for up to                |           |                                                                                                                                                                         |
| 540 | four years following an eruption, and resulted in positive precipitation anomalies over northwest (NW) and southeast                    |           |                                                                                                                                                                         |
|     | (SE) Australia. Larger atmospheric sulfate loading during larger volcanic eruptions coincided with more persistent               |           | Deleted: s                                                                                                                                                              |
| -   | positive IOD and El Nino conditions, enhanced positive precipitation anomalies over NW Australia, and dampened                          |           |                                                                                                                                                                         |
|     | precipitation anomalies over SE Australia.                                                                                              |           |                                                                                                                                                                         |
| 545 |                                                                                                                                         |           |                                                                                                                                                                         |
| 545 | 1. Introduction                                                                                                                         |           |                                                                                                                                                                         |
|     |                                                                                                                                         |           |                                                                                                                                                                         |
|     | Volcanic eruptions have significant impacts on weather and climate variability through the injection of volcanogenic                    |           |                                                                                                                                                                         |
|     | material into the atmosphere. Sulfate aerosols, formed through the reaction of $SO_2$ and $OH^-$ in the volcanic cloud,                 |           |                                                                                                                                                                         |
| 550 | decrease incoming shortwave radiation, and if injected into the stratosphere, can generate a global response (Driscoll et               |           |                                                                                                                                                                         |
|     | al., 2012; LeGrande et al., 2016). Previous studies have identified relationships between volcanism and surface and                     |           |                                                                                                                                                                         |
|     | tropospheric cooling (Driscoll et al., 2012), local stratospheric warming (Wielicki et al., 2002), strengthening of the                 |           |                                                                                                                                                                         |
|     | Arctic Oscillation and Atlantic meridional overturning circulation (Oman et al., 2005; Stenchikov et al., 2006, 2009 &                  |           |                                                                                                                                                                         |
|     | Shindell et al., 2004), and negative global precipitation anomalies (Gillet et al., 2004, Iles et al., 2013). The present               |           |                                                                                                                                                                         |
| 555 | study focuses on the under-studied relationship between large, globally significant tropical eruptions in the Last                      |           |                                                                                                                                                                         |
| 1   | Millennium (850-1850CE) and Australian precipitation through examination of the direct radiative aerosol effect and                     | A         | Deleted: ,                                                                                                                                                              |
|     | the feedbacks of two tropical modes that strongly influence Australian rainfall: the El-Nino Southern Oscillation                       | $\square$ | Deleted: and Australian precipitation                                                                                                                                   |
|     | (ENSO) and Indian Ocean Dipole (IOD),                                                                                                   | K A       | Formatted: Font:10 pt                                                                                                                                                   |
| •   |                                                                                                                                         | IA        | ENSO to volcanic eruptions.                                                                                                                                             |

[revised manuscript text omitted]

| Deleted: 6                               | 5                                                                                                                                                                 |
|------------------------------------------|-------------------------------------------------------------------------------------------------------------------------------------------------------------------|
| Deleted: 7                               | ,                                                                                                                                                                 |
| Deleted: S                               | substantial p                                                                                                                                                     |
| Deleted: v                               | with                                                                                                                                                              |
| Deleted: a                               | deficit                                                                                                                                                           |
| Deleted: b                               | )                                                                                                                                                                 |
| Deleted: r                               | ing                                                                                                                                                               |
| Deleted: a                               | and                                                                                                                                                               |
| Deleted:                                 | western Pacific and                                                                                                                                               |
| Deleted: s                               |                                                                                                                                                                   |
| Deleted: ,                               |                                                                                                                                                                   |
| Deleted: T
response th
eruption, w | he northern polar area (60-90°N) exhibits a greater
aan the south pole in the immediate years following
vith moderate cooling anomalies occurring over
s |

|     | Ensembles with volcanic forcing showed an increase in precipitation over southeast (SE) (Fig. 11) and northwestern                                                                                                                                                                                                                                                                                                                                                       | Deleted: 0 |
|-----|--------------------------------------------------------------------------------------------------------------------------------------------------------------------------------------------------------------------------------------------------------------------------------------------------------------------------------------------------------------------------------------------------------------------------------------------------------------------------|------------|
|     | (NW) (Fig. 2) Australia between July to November (JASON). Both areas showed predominantly positive anomalies in                                                                                                                                                                                                                                                                                                                                                          | Deleted: 8 |
|     | years 0-5 post-eruption, with the largest response seen between years 0-2. NW Australia (Fig. 9) showed larger positive                                                                                                                                                                                                                                                                                                                                                  | Deleted: 8 |
| 795 | precipitation anomalies between years 0-2 than SE Australia (Fig. 11) in the CR ensemble mean, and in years 0 and 2 in                                                                                                                                                                                                                                                                                                                                                   | Deleted: 0 |
| 1   | the 2xG ensemble mean.                                                                                                                                                                                                                                                                                                                                                                                                                                                   |            |
|     |                                                                                                                                                                                                                                                                                                                                                                                                                                                                          |            |
|     | Comparison of the precipitation anomalies following the Samalas and Huaynaptina eruptions in NW Australia (Fig. 10)                                                                                                                                                                                                                                                                                                                                                      | Deleted: 9 |
|     | showed that the smaller eruption had a delayed and smaller positive precipitation peak, with Samalas peaking in year 0                                                                                                                                                                                                                                                                                                                                                   |            |
| 800 | with an anomaly of 0.23 and Huaynaptina in year 2 at 0.14. While the Huaynaptina eruption also showed a delayed                                                                                                                                                                                                                                                                                                                                                          |            |
|     | peak in precipitation in SE Australia (Fig 12), the persistence of positive precipitation anomalies exceeded those of the                                                                                                                                                                                                                                                                                                                                                | Deleted: 1 |
|     | Samalas eruption. Huaynaptina recorded values > 0.17 in years 1-2 and a value of 0.12 in year 4, all of which were                                                                                                                                                                                                                                                                                                                                                       |            |
|     | larger anomalies than the peak of the Samalas eruption at 0.11 in year 1 (Fig. 12).                                                                                                                                                                                                                                                                                                                                                                                      | Deleted: 1 |
|     |                                                                                                                                                                                                                                                                                                                                                                                                                                                                          |            |
| 805 | Figures 13 and 14 show multi-volcano mean anomalous changes to the surface wind direction (Fig. 13) and speed (Fig.                                                                                                                                                                                                                                                                                                                                                      |            |
|     | 14) over the 5 years following eruptions in the CR forcing group. The most notable changes occur in years 0 and +1                                                                                                                                                                                                                                                                                                                                                       |            |
|     | where anomalously strong Southern Hemisphere westerlies and anomalously weak south easterly trade winds occurred,                                                                                                                                                                                                                                                                                                                                                        |            |
|     | accompanied by strong north-easterlies off the south-east coast of China and an intensification of North Atlantic                                                                                                                                                                                                                                                                                                                                                        |            |
|     | circulation. In year +3 anomalous south easterly winds off the south-western coast, and anomalous westerlies off the                                                                                                                                                                                                                                                                                                                                                     |            |
| 810 | central western coast, of South America are seen.                                                                                                                                                                                                                                                                                                                                                                                                                        |            |
|     | 4. Discussion and conclusions                                                                                                                                                                                                                                                                                                                                                                                                                                            |            |
|     | Our results suggest that the large scale IOD and ENSO sustame, and Australian rainfall regimes, were all imported by                                                                                                                                                                                                                                                                                                                                                     |            |
| 815 | but results suggest that the range-scale rolb and Errso systems, and Australian raintain regimes, were an impacted by                                                                                                                                                                                                                                                                                                                                                    |            |
| 015 | large tropical cruptions of the Last Wittennium.                                                                                                                                                                                                                                                                                                                                                                                                                         |            |

[revised manuscript text omitted]
|      | 30(15), doi: 10.1029/2003GL017926, 2003                                                                                                                                                                                                                                         |                         | Deleted: Individual and combined influences of ENSO and            |
|      |                                                                                                                                                                                                                                                                                 | 1 M                     | the Indian Ocean Dipole on the Indian summer monsoon               |

[revised manuscript text omitted]
 2012.                                                                                   |

---

## Author Response (AR2)

**1)** Comments from referees:**

**Anonymous Referee #2**

- 5 The authors have addressed most of the comments. The revision has improved substantially from the original submission, especially by clarify the model performance and specific goal of this study. I would recommend acceptance for publication once the following issues are addressed:
  - 1. Line 623 Please clarify the definition of "(July-September JASON)"
- 10 2. Line 740-744 The claim of "a significant pIOD condition one year after major eruptions in all volcanically forced ensembles that persists until year 5" needs more solid demonstration. Currently in Figure 4, DMI anomaly in the CR ensembles barely touch the significant line at year +1, let alone the other years; DMI anomaly in the 2xG scenario is slightly higher but the forcing is larger than the reality. Please also verify the use of 0.6 standard deviation as the significant level. "Fig.2" referred to in line 742 should be "Fig. 3".
- Figure 5 Does "the mean of all volcanic ensembles" mean the average of four CR ensembles and one 2xG ensemble? I assume the forcing in 2xG is much larger than that in CR. Please discuss how does the forcing with different magnitudes affect your results.
  - 4. Line 864-876 I do not find the added explanation of the mechanism underlying El Nino response satisfying. For example, how does "a uniform reduction in temperature" generate the El Nino-like anomaly? Why was it most visible in year +4?
  - 5. Line 880-882 Please demonstrate in more detail how did the direct precipitation effect of volcanic aerosols override the impact of the IOD on Australian precipitation. Right now, the conclusion was based on the results found in previous studies without verification in this model. Which parameter represents the direct precipitation effect of volcanic aerosols in the GISS model? How was the override effect actually appear in this model results?

1

6. I found the use of "multi-model" misleading, since it is only based on one model.

30

35

40

45

50

20

**2) Author's Response:**

55 Dear Reviewers,

Thank you for taking the time to read and comment on our paper. Your reviews were very helpful and we have now made several important revisions in light of Reviewer 2's further recommendations.

**60 Anonymous Referee #2**

Line 623 – Please clarify the definition of "(July-September – JASON)"

This has been reworded for clarity to "(July to September; JASON)" at line 235

65

70

Line 740-744 – The claim of "a significant pIOD condition one year after major eruptions in all volcanically forced ensembles that persists until year 5" needs more solid demonstration. Currently in Figure 4, DMI anomaly in the CR ensembles barely touch the significant line at year +1, let alone the other years; DMI anomaly in the 2xG scenario is slightly higher but the forcing is larger than the reality. Please also verify the use of 0.6 standard deviation as the significant level. "Fig.2" referred to in line 742 should be "Fig. 3".

We thank the reviewer for their comment that highlights this result was unclear and have made changes accordingly. The results of "a significant pIOD condition" at line 349 is valid as we have specified that we are judging the

- 75 significance on both the 0.6 standard deviation and by comparing the volcanic ensembles to the 'None' ensembles in both the figure captions (lines 736, 750, 771, 785), and now also in our methods section, where we have also verified that we've used the 0.6 standard deviation as the significance level (lines 325-339).
- However, we have now changed the wording from "that persists until year 5" to instead say that it persists for 4 years, from year +1 to year +4 (line 350). Once again, we thank the reviewer for highlighting this.

Fig. 2 has also been corrected to Fig. 3 as advised at line 351.

85 Figure 5 – Does "the mean of all volcanic ensembles" mean the average of four CR ensembles and one 2xG ensemble?

The figure does show the mean of the four CR ensembles and one 2xG ensemble, this has been clarified in the figure captions for Fig. 5 (line 743), 7 (line 757), 10 (line 775) and 12 (line 791).

**90**

Line 864-876 - I do not find the added explanation of the mechanism underlying El Nino response satisfying. For example, how does "a uniform reduction in temperature" generate the El Nino-like anomaly? Why was it most visible in year +4?

95 This paper has focused on characterising these anomalies here. The exact mechanism causing the El Nino response to the uniform reduction in temperature is an area that we have flagged for future research and believe would warrant a paper of its own. We would gladly collaborate with the reviewer on future work to explore the physical mechanisms in greater depth.

100 We believe the El-Nino anomaly was most visible in year +4 due to the occurrence of anomalous winds off the western

coast of South America in year +3, which we have now clarified further (line 460).

Line 880-882 – Please demonstrate in more detail how did the direct precipitation effect of volcanic aerosols override the impact of the IOD on Australian precipitation. Right now, the conclusion was based on the results found in previous studies without verification in this model. Which parameter represents the direct precipitation effect of volcanic aerosols in the GISS model? How was the override effect actually appear in this model results?

We thank the reviewer for their comments here and believe this question was the result of a misunderstanding around the use of the word 'direct'. We have now reworded line 467 to "Our results suggest that the direct effect of volcanic aerosols on precipitation overrode the impact of the IOD".

I found the use of "multi-model" misleading, since it is only based on one model.

115 We have now changed all 'multi-model' references to 'multi-ensemble' (lines 454, 730, 734, 749, 765, 770, 784, 803 and 808)

3

120

130

125

135

**3) Changes to manuscript**

[revised manuscript text omitted]
., Voulgarnkis, A., Yao, M. and Zhang, J.: CMP5 historical simulations (1850-2012) with GISS ModelE J. J. Adv. Model. Earth Sys., 6, 411-478, doi: 10.1002/2013MS000266, 2015.</li> <li>Moise, A., Bhend, H., Watteson, I. and Wilton, L.: Chapter 3: Evaluation of Climate Models. In: Climate Change in Australia Information for Australia Natural Resource Multipuope 3: Evaluation action of Climate Datasets. J. 10, 10102 (2013) Association of Australia Association of Australia Association. Climate Climate Report (CSIRO and Bureau of Meteorology), CSIRO and Bureau of Meteorology, Australia, 2015.</li> <li>Oman, L., Roboek, A., Stenchikov, G., Schmidt, G. A., and Ruedy, R.: Climatic response to high-latitude volcanic eruptions on ENSO and AMOC, PNAS, 112-45, 1378-13788, doi:10.1073/pnas.1509151112/2015.</li> <li>Pepier, A., Timbal, B., Rakich, C. and Courts-Smith, A.: Indian Ocean Dipole overrides ENSO's influence on cool season rainfall across the Eastern Schood on Australia. J Climat. 7, 3186-3262, doi: 10.1173/CLI-D13-005541, 2014.</li> <li>Peodybaylo, E., Schneikev, G. L., Wittenberg, A. T., and Zang, F.: Impacts of a Pinatubo-size volcanic eruption on ENSO, J Geophys. Res. Atmos, 122, 925-947, doi:10.1002/2016/D022706,2017.</li> <li>Robock, A.: Volcanic eruptions and climatr, Review Goophys. 38, 101-219, doi: 8755-1209/001/998RG000054, 2000.</li> <li>Robock, A. J. Uni, Y.J. Zubold, and L. Hu, and Navara, A.: Seasonality in the relationship between El Nino and the Indian Ocean dipole Clim. 7, 343-45, 2014.</li> <li>Kesy, M., Gualdi, S., Dthollav, H. L., and Navara, A.: Seasonality in the relationship between El Nino and the Indian Ocean dipole Clim. Dynam, 37, 221-256, doi: 10.1075/012014/010976-12014/0110976-12014/0110976-10020130000265, 2014.</li> <li>Stato, H., Hansen, J. E., McCormick, M. P., and Pollack, J. B.: Statospheric aerosol optical depths, 1830-1990, J. Geophys. Res., 98, 22897-299</li>                      |     |                                                                                                                                                                                                                                                                                                                                                                                                      |
|------------------------------------------------------------------------------------------------------------------------------------------------------------------------------------------------------------------------------------------------------------------------------------------------------------------------------------------------------------------------------------------------------------------------------------------------------------------------------------------------------------------------------------------------------------------------------------------------------------------------------------------------------------------------------------------------------------------------------------------------------------------------------------------------------------------------------------------------------------------------------------------------------------------------------------------------------------------------------------------------------------------------------------------------------------------------------------------------------------------------------------------------------------------------------------------------------------------------------------------------------------------------------------------------------------------------------------------------------------------------------------------------------------------------------------------------------------------------------------------------------------------------------------------------------------------------------------------------------------------------------------------------------------------------------------------------------------------------------------------------------------------------------------------------------------------------------------------------------------------------------------------------------------------------------------------------------------------------------------------------------------------------------------------------------------------------------------------------------------------------------------------|-----|------------------------------------------------------------------------------------------------------------------------------------------------------------------------------------------------------------------------------------------------------------------------------------------------------------------------------------------------------------------------------------------------------|
|  <li>Moise, A., Bhend, H., Watterson, I. and Wilson, L.: Chapter 5: Evoluation of Climate Models. In: Climate Change in Australia
Information for Australia's Natural Resource Management Regions: Technical Report [CSIRO and Bureau of Meteorology]. CSIRO
and Bureau of Meteorology, Australia, 2015.</li> <li>Goophys, Res, 110, D13103, doi:10.1029/2004JD005487, 2005.</li> <li>Paustar, F. S. R., Chaff, L., Caballero, R., and Rattisti, D. S.: Impact of high-latitude volcanic eruptions on ENSO and AMOC,
PNAS, 112-45, 13784-13788, doi:10.1073/pnas.1509153112, 2015.</li> <li>Pepter, A., Timbal, B., Rakich, C. and Coutts-Smith, A.: Indian Ocean Dipole overrides ENSO's influence on cool season rainfall
across the Eastern Seaboard of Australia, J Clim., 27, 3816-3826, doi: 10.1175/JCLI-D-13-005541, 2014.</li> <li>Predybaylo, E., Stenchikov, G. L., Wittenberg, A. T., and Zeng, F.: Impacts of a Pinatubo-size volcanic eruption on ENSO, J
Goophys, Res. Atmos., 122, 925-947, doi:10.1002/2016/D025796, 2017.</li> <li>Robock, A.: Volcanic eruptions and climate, Reviews Geophys., 38, 191-219, doi: 8755-1209/00/1998RG000054, 2000.</li> <li>Robock, A.: Volcanic eruptions and climate, Reviews Geophys., 38, 191-219, doi: 8755-1209/00/1998RG000054, 2000.</li> <li>Robock, A. and Liu, Y.: The volcanic signal in Goddard Institute for Space Studies three-dimensional model simulations, J Clim., 7,
44-55, doi: 10.1175/JCLIA-014(1994)007-004471 YSIG1>-20.02, 1994.</li> <li>Ruston, G.D.: The unequal virtuance t-test is underticed alternative to Student's Letst and the Mann-Whitney U test, Behavioral
Ecology 4, 688-690, doi: 10.1093/behecourt016.2006,</li> <li>Satio, N.H., Xie, SP., and Yamaggata, T: Tropical Indian Ocean Variability in the IPCC Twentieth-Century Climate Simulations, J.
Clim., 19, 497-4417, doi: 10.1073/JCLIA-31, 2006.</li> <li>Sato, M., Hansen, J. E., McCormick, M. P., and Pollack, J. B.: Stratospheric aerosol optical depths, 1850-1990, J. Geophys, Res., 98,
22987-22984, doi:10</li>   |     | D., Romanou, A., Russel, G.L., Sato, M., Sun, S., Tsigaridis, K., Unger, N., Voulgarakis, A., Yao, M. and Zhang, J.: CMIP5 historical simulations (1850-2012) with GISS ModelE2, J. Adv. Model. Earth Sys., 6, 441-478, doi: 10.1002/2013MS000266, 2015.                                                                                                                                             |
|  <li>Oman, L., Robock, A., Stenchikov, G., Schmidt, G. A., and Ruedy, R.: Climatic response to high-latitude volcanic eruptions, J Grophys. Res., 110, D13103, doi:10.1029/2004/D005487, 2005.</li> <li>Pausata, F. S. R., Chaffk, L., Caballero, R., and Battisti, D. S.: Impact of high-latitude volcanic eruptions on ENSO and AMOC, PNAS, 11245, 13784-1358, doi:10.1073/msis.1509153112, 2015.</li> <li>Pepler, A., Timbal, B., Rakich, C. and Coutts-Smith, A.: Indian Ocean Dipole overrides ENSO's influence on cool season minfall across the Eastern Seaboard of Australia, J Chm., 27, 3816-3826, doi: 10.1173/JCL1P-D14005541, 2014.</li> <li>Predybaylo, E., Stenchikov, G. L., Wittenberg, A. T., and Zeng, F.: Impacts of a Pinatubo-size volcanic eruption on ENSO, J Grophys. Res. Atmos., 12, 953-947, doi:10.1022/016/D02569, 2017.</li> <li>Robock, A.: Volcanic eruptions and elimate, Reviews Geophys., 38, 191-219, doi: 8755-1209/00/1998RG000054, 2000.</li> <li>Robock, A.: Volcanic eruptions and elimate, Reviews Geophys., 38, 191-219, doi: 8755-1209/00/1998RG000054, 2000.</li> <li>Robock, A.: Volcanic eruptions and elimate, Reviews Geophys., 38, 191-219, doi: 8755-1209/00/1998RG000054, 2000.</li> <li>Robock, A.: Volcanic eruptions and elimate, Reviews Geophys., 38, 191-219, doi: 8755-1209/00/1998RG000054, 2000.</li> <li>Robock, A.: Guidi, S., Drobuby, H. L., and Naverra, A.: Seasonabili yin the relationship between El Nino and the Indian Ocean dipole Clim. Dynam., 37, 221-236, doi: 10.1007/s00382-010-0876-1, 2011.</li> <li>Ruxton, G.D.: The unequal variance-test is an underused alternative to Stduent's t-test and the Mann-Whitney U test, Behavioral Ecology, 4, 688-900, doi: 10.0137/bctClas4471, 2006.</li> <li>Sato, M., Hansen, J. E., McCormick, M. P., and Pollack, J. B.: Stratospheric aerosol optical depths, 1850-1990, J. Geophys. Res., 98, 22987-2294, doi:10.102993/D02531, 1993.</li> <li>Schmidt, G. A., Kelley, M., Nazarenko, L., Ruedy, R., Rassel, G., Aleinov, I., et al: Configur</li>                                                 | 615 | Moise, A., Bhend, H., Watterson, I. and Wilson, L.: Chapter 5: Evaluation of Climate Models . In: Climate Change in Australia
Information for Australia's Natural Resource Management Regions: Technical Report [CSIRO and Bureau of Meteorology]. CSIRO
and Bureau of Meteorology, Australia, 2015.                                                                                    |
|  <li>Pausata, F. S. R., Chaffk, L., Caballero, R., and Battisti, D. S.: Impact of high-latitude volcanic eruptions on ENSO and AMOC, PNAS, 11245, 11245, 11246, 11246, 11246, 11246, 11246, 11246, 11246, 11246, 11246, 11246, 11246, 11246, 11246, 11246, 11246, 11246, 11246, 11246, 11246, 11246, 11246, 11246, 11246, 11246, 11246, 11246, 11246, 11246, 11246, 11246, 11246, 11246, 11246, 11246, 11246, 11246, 11246, 11246, 11246, 11246, 11246, 11246, 11246, 11246, 11246, 11246, 11246, 11246, 11246, 11246, 11246, 11246, 11246, 11246, 11246, 11246, 11246, 11246, 11246, 11246, 11246, 11246, 11246, 11246, 11246, 11246, 11246, 11246, 11246, 11246, 11246, 11246, 11246, 11246, 11246, 11246, 11246, 11246, 11246, 11246, 11246, 11246, 11246, 11246, 11246, 11246, 11246, 11246, 11246, 11246, 11246, 11246, 11246, 11246, 11246, 11246, 11246, 11246, 11246, 11246, 11246, 11246, 11246, 11246, 11246, 11246, 11246, 11246, 11246, 11246, 11246, 11246, 11246, 11246, 11246, 11246, 11246, 11246, 11246, 11246, 11246, 11246, 11246, 11246, 11246, 11246, 11246, 11246, 11246, 11246, 11246, 11246, 11246, 11246, 11246, 11246, 11246, 11246, 11246, 11246, 11246, 11246, 11246, 11246, 11246, 11246, 11246, 11246, 11246, 11246, 11246, 11246, 11246, 11246, 11246, 11246, 11246, 11246, 11246, 11246, 11246, 11246, 11246, 11246, 11246, 11246, 11246, 11246, 11246, 11246, 11246, 11246, 11246, 11246, 11246, 11246, 11246, 11246, 11246, 11246, 11246, 11246, 11246, 11246, 11246, 11246, 11246, 11246, 11246, 11246, 11246, 11246, 11246, 11246, 11246, 11246, 11246, 11246, 11246, 11246, 11246, 11246, 11246, 11246, 11246, 11246, 11246, 11246, 11246, 11246, 11246, 11246, 11246, 11246, 11246, 11246, 11246, 11246, 11246, 11246, 11246, 11246, 11246, 11246, 11246, 11246, 11246, 11246, 11246, 11246, 11246, 11246, 11246, 11246, 11246, 11246, 11246, 11246, 11246, 11246, 11246, 11246, 11246, 11246, 11246, 11246, 11246, 11246, 11246, 11246, 11246, 11246, 11246, 11246, 11246, 11246, 11246, 11246, 11246, 11246, 11246, 11246, 11246, 11246, 11246, 11246, 11246, 11246, 11246, 11</li>     | 620 | Oman, L., Robock, A., Stenchikov, G., Schmidt, G. A., and Ruedy, R.: Climatic response to high-latitude volcanic eruptions, J Geophys. Res., 110, D13103, doi:10.1029/2004JD005487, 2005.                                                                                                                                                                                                            |
|  <li>Pepler, A., Timbal, B., Rakich, C. and Coutta-Smith, A.: Indian Ocean Dipole overrides ENSO's influence on cool season rainfall across the Eastern Seaboard of Australia, J Clim., 27, 3816-3826, doi: 10.1175/JCLLD-13-00554.1, 2014.</li> <li>Predybaylo, E., Stenchikov, G. L., Wittenberg, A. T., and Zeng, F.: Impacts of a Pinatubo-size volcanic eruption on ENSO, J Geophys, Res. Atmos., 12, 295-547, doi:10.1002/2016J025796, 2017.</li> <li>Robock, A.: Volcanic eruptions and climate, Reviews Geophys., 38, 191-219, doi: 8755-1209/00/1998RG000054, 2000.</li> <li>Robock A and Liu, Y.: The volcanic signal in Goddard Institute for Space Studies three-dimensional model simulations, J Clim., 7, 44–55, doi: 10.1175/JS200442(1994)007-0044.TVSIGD-2.0.CO2, 1994.</li> <li>Rocy, M., Gualdi, S., Drobhlav, H. L., and Navarra, A.: Seasonality in the relationship between El Nino and the Indian Ocean dipole Clim. Dynam. 37, 221-236, doi: 10.1007/s00382-010-04876-1, 2011.</li> <li>Ruxton, G.D.: The unequal variance t-test is an underused alternative to Stduent's t-test and the Mann-Whitney U test, Behavioral Ecology, 4, 688-690, doi: 10.1009/Jshchccdardolf 6, 2006.</li> <li>Saji, N. H., Xie, SP. and Yamagat, T.: Topical Indian Ocean Variability in the IPCC Twentieth-Century Climate Simulations, J. Clim., 19, 4397-4417, doi:10.1175/JCL13847.1, 2006.</li> <li>Sato, M., Hansen, J. E., McCornick, M. P., and Pollack, J. B.: Stratospheric acrosol optical depths, 1850-1990, J. Geophys. Res., 98, 22987-22994, doi:10.1029/3JD02553, 1993.</li> <li>Schmidt, G. A., Kelley, M., Nazareko, L., Ruedy, R., Russel, G., Alcinov, I., et al.: Configuration and assessment of the GISS ModelE2 contributions to the CMIPS archive, J. Advances in Modeling Earth Systems, 6:1, 141-184, doi: 10.1002/2013MS000265, 2014.</li> <li>Schmidt, G. A., Kelley, M., Nazareko, L., Ruedy, R., Russel, G., Jleinov, I., et al.: Configuration and assessment of the GISS ModelE2 contributions to the CMIPS archive, J. Advances</li>                       |     | Pausata, F. S. R., Chafik, L., Caballero, R., and Battisti, D. S.: Impact of high-latitude volcanic eruptions on ENSO and AMOC, PNAS, 112:45, 13784-13788, doi:10.1073/pnas.1509153112, 2015.                                                                                                                                                                                                        |
|  <li>Predybaylo, E., Stenchikov, G. L., Wittenberg, A. T., and Zeng, F.: Impacts of a Pinatubo-size volcanic eruption on ENSO, J Geophys. Res. Atmos., 122, 925-947, doi:10.1002/2016JD023766, 2017.</li> <li>Robock, A.: Volcanic eruptions and climate, Reviews Geophys., 38, 191-219, doi: 8755-1209/00/1998RG000054, 2000.</li> <li>Robock A and Liu, Y.: The volcanic signal in Goddard Institute for Space Studies three-dimensional model simulations, J Clim., 7, 44–55, doi: 10.1175/15200442(1994)007-0044-1YS015-2012.0202, 1994.</li> <li>Roxy, M., Gualdi, S., Drbohlav, H. L., and Navarra, A.: Seasonality in the relationship between El Nino and the Indian Ocean dipole Clim. Dynam., 37, 221-236, doi: 10.1007/s00382-010-0876-1, 2011.</li> <li>Ruxton, G.D.: The unequal variance t-test is an underused alternative to Stduent's t-test and the Mann-Whitney U test, Behavioral Ecodogy, 4, 688-00, doi: 10.1029/artica4016.2006, 2006.</li> <li>Saji, N. H., Xie, SP., and Yamagata, T.: Tropical Indian Ocean Variability in the IPCC Twentieth-Century Climate Simulations, J. Clim., 19, 4397-4417, doi:10.1175/JCL13847.1, 2006.</li> <li>Sato, M., Hansen, J. E., McCorrnick, M. P., and Pollack, J. B.: Stratospheric aerosol optical depths, 1850-1990, J. Geophys. Res., 98, 2287-2294, doi:10.1029/93JD02553, 1993.</li> <li>Schmidt, G. A., Kelley, M., Nazarenko, L., Ruedy, R., Russel, G., Aleinov, I., et al.: Configuration and assessment of the GISS ModelE2 contributions to the CMIPS archive, J. Advances in Modeling Earth Systems, 6.1, 141-184, doi: 10.1002/2008JD011222, 2009.</li> <li>Self, S., Rampino, M. R., Zhao, J., and Katz, M. G.: Volcanic aerosol perturbations and strong El Nino events: No general correlation, Geophys Res Lett, 24, 1247-1250, 1997.</li> <li>Shindell, D., Schmidt, G., Mann, M., and Faluvegi, G.: Dynamic winter climate response to large tropical volcanic cruptions since 1600, J Geophys. Res., 109, D05104, doi:10.1029/2003JD00451, 2004.</li> <li>Sigl, M., McConnell, J. R., Lay</li>                                               |     | 
[revised manuscript text omitted]